# Skills-in-Context Prompting: Unlocking Compositionality in Large Language Models

## Abstract

We consider the problem of eliciting compositional generalization capabilities in large language models (LLMs) with a novel type of prompting strategy. Compositional generalization empowers the LLMs to solve problems that are harder than the ones they have seen (i.e., easy-to-hard generalization), which is a critical reasoning capability of human-like intelligence. However, even the current state-of-the-art LLMs still struggle with this form of reasoning. To bridge this gap, we propose *skills-in-context* (SKiC) prompting, which instructs LLMs how to compose basic skills to resolve more complex problems. We find that it is crucial to demonstrate both the skills and the compositional examples within the same prompting context. With as few as two examplars, our SKiC prompting initiates strong synergies between skills and their composition capabilities. Notably, it empowers LLMs to solve unseen problems that require innovative skill compositions, achieving near-perfect generalization on a broad range of challenging compositionality tasks. Intriguingly, SKiC prompting unlocks the latent potential of LLMs, enabling them to leverage pre-existing internal skills acquired during earlier pretraining stages, even when these skills are not explicitly presented in the prompting context. This results in the capability of LLMs to solve unseen complex reasoning problems by activating and composing internal competencies. With such prominent features, SKiC prompting is able to significantly improve the mathematical reasoning capabilities (e.g., MATH) with simple 1-stage prompting.

## 1 Introduction

Large language models (LLMs) have achieved great success in solving natural language processing (NLP) tasks (Brown et al., 2020; Radford et al., 2019; Smith et al., 2022; Chowdhery et al., 2022; Lewkowycz et al., 2022; Sanh et al., 2021; Wei et al., 2021; Mishra et al., 2022; Chung et al., 2022; Ouyang et al., 2022; OpenAI, 2023; Touvron et al., 2023b). When the size of the model scales up, LLMs exhibit strong zero-shot and few-shot performance on a wide range of NLP tasks — a salient behavior characterized by the scaling law (Kaplan et al., 2020; Hoffmann et al., 2022) and emergent abilities (Wei et al., 2022a). However, LLMs still struggle with compositional generalization, i.e., the ability to use existing skills to solve more complex unseen problems (i.e., "easy-to-hard" generalization) (Zhou et al., 2022; Dziri et al., 2023). In this paper, we will develop a prompting strategy for LLMs that can effectively unlock their generic compositional generalization capabilities.

Ideally, if an LLM has already learned a rich set of skills, it should be able to solve any problem whose solutions are composable from these skills. To unlock such great potential, the key is to teach the LLMs how to use these skills to construct a solution to any unseen, more difficult problem. Towards this goal, there have been a series of prompting strategies being developed to improve the reasoning and compositionality capabilities. Notably, chain-of-thought (CoT) prompting (Wei et al., 2022b) significantly improves the reasoning performance of LLMs by demonstrating how to approach a complex problem through a sequence of simple and basic steps. Follow-ups such as Least-to-Most prompting (Zhou et al., 2022) and decomposed prompting (Khot et al., 2022) propose a two-stage strategy, which first decomposes the original problem into a set of subproblems, and then solve and combine each of them sequentially. Although these methods significantly boost the performance over standard prompting in solving many challenging compositional generalization tasks, they still cannot perform systematic generalization well enough over problems that are significantly harder than the ones they have seen. Moreover, least-to-most prompting and decomposed

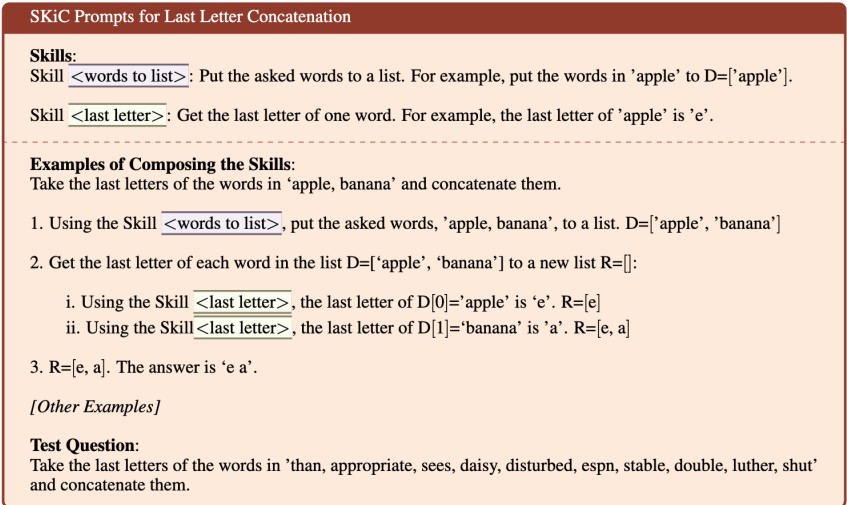

Figure 1: Skills-in-Context Prompting. The prompt consists of three blocks: (i) the (basic) skills for solving a complex task, (ii) (few-shot) examples of how to compose the skills, and (iii) the problem to be solved. The above prompt will be fed into an LLM to generate the output — see Figure 20 for an example of the output. Note that the compositional examplars demonstrate how to ground the reasoning steps onto the basic skills (highlighted in colors) for solving a complex problem.

prompting are restricted to solving a certain class of tasks, where each problem can be decomposed as a sequence of subproblems. And for problems with general computation graphs (Dziri et al., 2023), it is generally less intuitive, if not possible, to construct the prompting exemplars.

In this paper, we develop an effective *one-stage* prompting strategy, named **SK**ills-**i**n-**C**ontext (SKiC) prompting, to unlock the general compositional generalization capability in LLMs. The key insight is to teach the LLM to explicitly ground each of its reasoning steps on the (more elementary) skills. Specifically, the SKiC prompt is constructed from three main blocks. The first block contains a (non-exhaustive) list of skills that LLMs may need to use in order to solve a more complex problem, which include both descriptions of the skills and the instructions (with a few examples) on how to use them. These skills can be distilled either manually or automatically via prompting the LLMs. The second part consists of a few (generally two in most of our cases) examplars that demonstrate how to explicitly compose skills into a solution to a more complex problem. The last part is the problem to be solved. Interestingly, with both the skills and their explicit compositions presented in the context, the LLMs successfully learn how to ground each reasoning step on the knowledge and skills that they have already mastered, yielding the desirable general compositional generalization capabilities. Notably, unlike the Least-to-Most or decomposed prompting, our proposed approach is a one-stage prompting method, without the need to call LLMs multiple times. Therefore, it can be easily used in a plug-and-play manner, as the CoT prompting and the standard prompting.

We evaluate our proposed SKiC prompting on a wide range of challenging compositional generalization tasks. Our experiments show that SKiC prompting achieves state-of-the-art performance on all of these benchmarks, and it even achieves near-perfect generalization on unseen harder problems on some of the datasets. Moreover, the improvement margins compared to the previous methods are significant. For example, SKiC outperforms previous state-of-the-art prompting strategies on unseen longer cases by 16.3% on last-letters (Zhou et al., 2022), 25.5% on addition, 45.0% on multiplication (Dziri et al., 2023),9.93% on Commaqa-E (Khot et al., 2021), 36.0% on dynamic programming (Dziri et al., 2023), 2.7% on GSM8K(Cobbe et al., 2021), and 12.1% on MATH (Hendrycks et al., 2021). Notably, our results on GSM8K and MATH further reveal that SKiC prompting allows the LLMs to generalize beyond the skills provided in the context and solve problems by using the vast reservoir of the internal skills they acquired during the prior pretraining stage. It clearly demonstrates that SKiC prompting unleashes strong synergies between skills and their composition capabilities, which teaches LLMs to generalize to harder problems than they have seen and to problems that require innovative compositions of existing knowledge (either in context or inside model weights).

## 2 METHODOLOGY

While humans naturally exhibit compositional generalization in problem-solving, LLMs often struggle to compose *basic skills* in innovative ways to solve more difficult problems (Dziri et al., 2023). Empowering LLMs with the ability to compose the skills that they have seen to solve more complex tasks is important to mirror human intelligence and to reach superintelligence. To this end, we introduce a novel prompt methodology, Skills-in-Context (SKiC) prompting, to teach language models composing elementary skills to solve problems for better compositional generalization.

### 2.1 SKILLS-IN-CONTEXT PROMPTING

Skills-in-context prompting facilitates compositional generalization by explicitly instructing language models to utilize basic skills to solve complex problems.[1] This aligns with the success of Elaborative Rehearsal in the human learning process (Berry, 1983; McNamara & Magliano, 2009), where studies (Kheirzadeh & Pakzadian, 2016) have demonstrated that by first summarizing relevant knowledge and skills as the Scaffolding (Hammond & Gibbons, 2005) and establishing connections between the problem-solving steps and the existing Scaffolding, human would process the new information with greater depth and thoroughness, thus reinforcing both the concepts and their practical applications. Inspired by such human cognition theory (Berry, 1983; Bakker et al., 2015), a SKiC prompt consists of three major parts: **(i)** Characterization of the basic skills that are needed to solve complex problems, including the description of the skills and the instruction on how to use them (with few-shot examplars). **(ii)** Examples of how to compose basic skills into solutions to complex problems. **(iii)** The problem to be solved. An example is shown in Figure 1. The language model is first provided with several skills (*scaffolding*) such as getting the last letter of one word followed by several examples introduced to illustrate the process of utilizing these basic skills to answer the complex problem (*elaborative rehearsal*). For example, to take the last letter of a series of words, language models use the "words_to_list" skill to add the words to a list and then use the "last_letter" skill to iteratively obtain the last letter of each word.

**Comparison to previous prompting strategies** Figure 5 (in Appendix) visualizes the differences between our proposed SKiC prompting and the previous related prompting methods. Different from Chain-of-Thoughts prompting, our SKiC prompting provides explicit grounding on the basic skills for reasoning steps towards final answers inspired by elaborative rehearsal(Berry, 1983). Compared to recent prompting methods for handling compositional problems such as Least-to-Most prompting (LtM) (Zhou et al., 2022) and Decomp (Khot et al., 2022), our SKiC is superior in several dimensions: (i) Our SKiC prompting is more general to solve extended sets of problems. Previous decomposing-based approaches like LtM and Decomp usually solve complex problems in a two-stage fashion by first decomposing the problem into a linear sequence of subproblems and then solving them sequentially. However, many of the tasks that have complex computation graphs such as multiplication and dynamic programming problems (Dziri et al., 2023) cannot be decomposed in a simple manner, which makes it hard to apply these decomposition-based approaches. (ii) The decomposition operation can also be viewed as one basic skill in our SKiC prompt (for example, we view the decomposition operation as one of the skills in the question-answer task in Figure 12). (iii) SKiC solves the complex problems in a single stage, which could alleviate the error propagation compared to decomposition-based approaches that require multiple distinct stages. Due to the one-stage nature, our SKiC prompting can replace other one-stage strategies such as the CoT promptings in a plug-and-play manner. And it can also be easily combined with other ensemble techniques such as self-consistency (Wang et al., 2022) and Progressive-Hint Prompting (Zheng et al., 2023) to further boost the performance.

### 2.2 CONSTRUCTION OF THE SKIC PROMPTS

One key step in constructing our SKiC prompts is to distill the (basic) skills that might be needed for solving problems associated with a task. We now introduce two approaches (shown in Figure 2).

**Distill Skills via Human** This is a fully manual approach, where the basic skills are manually summarized from a few (less than 10) problems. For example, given several samples from the last-letter-

---

[1]The term "basic skills" within SKiC prompting are not necessarily atomic skills. Rather, they could be any skills (e.g., a composite skill by itself) that serve as the foundational blocks for tackling complex problems.

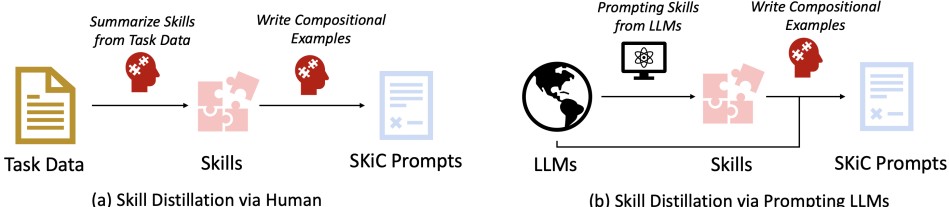

Figure 2: Two approaches to creating SKiC prompts, depending on how we distill the skills. (a) We *manually* summarize the skills from the sample problems, and then construct the compositional examplars on how to compose these skills. (b) We prompt the LLMs to *automatically* generate the necessary skills, followed by human review. Then we manually craft the compositionl examplars by grounding their reasoning steps onto either the provided in-context skills or the inherent skills within the LLMs, where the existence of the inherent skills is verified by zero-shot prompting.

concatenation task, we manually identify that "words_to_list" and "last_letter" are common skills to be used. Based on the discovered skills, we add a few ($1 \sim 2$) simple examples to illustrate these basic skills alone. Once the in-context skills are constructed, we add the compositional examplars to demonstrate the composition of these skills to solve a problem (Figure 1). This approach puts all the essential skills in the context and is generally applicable to narrow domain problems that require the composition of limited skills for solving harder problems. It is also beneficial for semi-parametric LLMs, which can dynamically access the most relevant skills from external memories based on each input instance and integrate them into the problem context (Borgeaud et al., 2022; Pan et al., 2022; Izacard et al., 2022).

**Distill Skills via Prompting LLMs** This is a semi-automatic approach, where we first prompt the LLMs to automatically generate the necessary basic skills (i.e., the descriptions and examples) followed by human review. For instance, when identifying the skills required to address the mathematical problems in the MATH task (Hendrycks et al., 2021), we prompt the LLM with phrases like "basic skills in Algebra". This leads the model to generate foundational skills, such as "Factoring" (see Figure 18 for the full list of the skills). Next, we manually construct the compositional examplars by grounding the reasoning steps on the skills. It is worth noting that an exemplar might require skills not explicitly presented in the prompt context. In these instances, we anchor the reasoning to inherent skills within the LLMs, confirming their presence through zero-shot prompting. For example, in the compositional exemplar showcased in Figure 19, aside from leveraging in-context skills such as "Combination" and "Sub", it also employs skills like "Pascal's Triangle" — a capability not present in the context but inherently known to the LLM. Such a construction of the examplars will encourage the model to generalize beyond the in-context skills and compose solutions from the internal capabilities as well — see Figure 33 for an example of the generated solution that activates the internal skills <Angle Bisector Theorem> and <Heron's Formula>. To be more specific, for every problem in the MATH task, around 24% of the skills, as shown in Table 5, applied in the reasoning steps stem from the LLM's internal pre-trained knowledge (see Table 12 for the most frequently used internal skills). The ability to harness both in-context skills and inherent capabilities is crucial for addressing complex reasoning problems, which typically require varied compositions across a broad spectrum of foundational skills. Manually enumerating every required skill within a prompt context is often impractical. Meanwhile, LLMs have accumulated a vast reservoir of knowledge and skills during their pre-training. Leveraging these internal competencies can unlock significant potential, allowing LLMs to tackle even more complex challenges.

# 3 EXPERIMENTS

## 3.1 EXPERIMENTAL SETTINGS

In our experiments, we will show the superior compositional capabilities of SKiC prompting by evaluating in two broad settings. Details of each task can be found in the Appendix B:

**Composition over in-context skills**, where all the essential skills needed to solve the problems are provided in the context. The tasks we evaluated in this setting include (i) symbolic manipulation(last

letter concatenation) (Wei et al., 2022b; Zhou et al., 2022; Khot et al., 2022), where the LLM needs to generate the concatenation of the last letter from each word in a given list of $1 \sim 12$ words, (ii) arithmetic operation (addition and multiplication) (Dziri et al., 2023), where the LLM needs to generate the sum of two number with $2\sim7$ digits and the product of two numbers with $2\sim5$ digits, (iii) question answering (CommaQA-E) (Khot et al., 2022), where for given facts of entities, the models need to answer the multi-hop questions, e.g., *What movies have people from the country Stridery acted in?*, and (iv) dynamic programming (Dziri et al., 2023), where for a sequence of $4\sim8$ integers, LLMs need to find the highest sum for a subsequence where no two numbers are adjacent. These tasks require only a limited skill set, yet pose challenges in terms of easy-to-hard generalization capabilities. Under these circumstances, we construct our SKiC prompts manually, adhering to the first methodology outlined in Section 2.2. The skills and examples of how to compose these skills are shown in Figures 6-7 for last letter concatenation, Figures 8-9 for addition, Figures 10-11 for multiplication, Figures 12-13 for CommaQA-E, and Figures 14-15 for dynamic programming.

**Generalization beyond in-context skills**, where models also need to harness skills beyond what have been provided in the context and tap into the internal skills for math reasoning like GSM8K (Wei et al., 2022b; Zhou et al., 2022) and MATH (Hendrycks et al., 2021) problems. In this context, the primary challenge lies in achieving diverse compositions across a wide range of skills to solve a complex reasoning problem. Thus, we evaluate whether our SKiC prompting could allow LLMs to invoke the massive set of internal skills and knowledge that are acquired during pre-training. Such capability is vital in solving complex reasoning problems (e.g., math), which require varied compositions over a vast amount of foundational skills. And it is impractical to enumerate all the skills in context. For GSM8K (Cobbe et al., 2021), which is a set of relatively simpler math problems that could mostly be solved by simple math operations such as multiplication, addition and etc. As a result, we construct SKiC following the first approach as described in Section 2.2. The skill sets and one example of how to compose these skills to solve a GSM8K problem are shown in Figure 16 and Figure 17, respectively. For MATH (Hendrycks et al., 2021), which is a significantly more challenging benchmark on mathematical reasoning with problems in Algebra, Counting and Probability, Geometry, Intermediate Algebra, Number Theory, PreAlgebra, and PreCalculus. It is infeasible to distill and enumerate the needed skills manually. Therefore, we adopt the second approach as described in Section 2.2, where we prompt the LLMs to generate the skills and then craft the compositional examples manually shown in Figures 18-19.

For all the experiments, we mainly compare our SKiC with zero/few-shot standard prompting (4-shot) (Brown et al., 2020), CoT (Wei et al., 2022b), Least-to-Most prompting (LtM) (Zhou et al., 2022) and Decomp prompting (Khot et al., 2022) [2] on different foundation models including LLAMA-65B (Touvron et al., 2023a), text-davinvi-003 (Brown et al., 2020), ChatGPT and GPT4 (OpenAI, 2023). For tasks in the second setting, we further compare our methods with Scratchpad (Nye et al., 2021), Learning-to-Program(LtP) (Guo et al., 2023), and ComplexCoT (Fu et al., 2022) as well as different ensemble strategies that are commonly combined together with these baselines: majority voting (maj1@k) (Lewkowycz et al., 2022), Self-Consistency (SC) (Wang et al., 2022), and Progressive-Hint Prompting (PHP) (Zheng et al., 2023). Note that all the compositional exemplars in SKiC prompts are either a subset of or the same as what have been used in baselines. Additional experiments on LLAMA2 (Touvron et al., 2023b) can be found in Appendix C.

## 3.2 MAIN RESULTS

**Composition over In-Context Skills: Easy-to-Hard Generalization** We report the evaluation results for last letter concatenation, addition&multiplication, Commmaqa-E and DP in Tables 1-4. We observe that standard zero/few-shot prompting generalizes poorly on the problems that are harder than the examplars in the prompting context. For example, on last letter concatenation tasks, 4-shot standard prompting only achieves 10% accuracy with text-davinci-003 when solving testing problems that involve 12 words. CoT, LtM and Decomp prompting improve the overall performance but still degrade quickly over longer inputs (e.g., CoT slightly improves the accuracy on arithmetic tasks, LtM outperform CoT on last letter concatenation and Decomp prompting boosts the exact match on Commaqa-E dataset.) Our Skills-in-Context prompting significantly boosts the performance with less demonstration examples on all the tasks especially in harder cases (e.g., gaining over

---

[2]We exclude the LtM or Decomp prompting in arithmetic and dynamic programming tasks as it is difficult to design linear problem decomposition in these settings.

Table 1: Accuracy on different evaluation subsets of the last-letter-concatenation task. The testing problems with 1 and 2 words are in-distribution evaluation, while the ones with $4 \sim 12$ words are (harder) out-of-distribution evaluations.

| Model | Prompting | #-shots | 1 | 2 | 4 | 6 | 8 | 10 | 12 |
|---|---|---|---|---|---|---|---|---|---|
| LLAMA-65B | zero-shot | 0 | 0 | 0 | 0 | 0 | 0 | 0 | 0 |
| | 4-shots | 4 | 72.0 | 66.0 | 50.0 | 26.0 | 10.0 | 6.0 | 0 |
| | CoT | 4 | 76.0 | 70.0 | 58.0 | 42.0 | 30.0 | 26.0 | 20.0 |
| | LtM | 4 | 76.0 | 72.0 | 66.0 | 50.0 | 46.0 | 36.0 | 25.0 |
| | SKiC | 2 | **81.0** | **97.0** | **77.0** | **59.0** | **56.0** | **48.0** | **36.0** |
| text-davinci-003 | zero-shot | 0 | 0 | 0 | 0 | 0 | 0 | 0 | 0 |
| | 4-shots | 4 | 99.0 | 97.0 | 89.0 | 68.0 | 45.0 | 27.0 | 10.0 |
| | CoT | 4 | 100.0 | 99.0 | 90.0 | 75.0 | 52.0 | 39.0 | 31.0 |
| | LtM | 4 | 100.0 | 99.0 | 94.0 | 90.0 | 87.0 | 84.0 | 80.0 |
| | SKiC | 2 | **100.0** | **100.0** | **100.0** | **100.0** | **100.0** | **99.0** | **98.0** |
| ChatGPT | zero-shot | 0 | 99.0 | 98.0 | 93.0 | 88.0 | 84.0 | 80.0 | 77.0 |
| | 4-shots | 4 | 100.0 | 100.0 | 95.0 | 92.0 | 90.0 | 86.0 | 85.0 |
| | CoT | 4 | 100.0 | 100.0 | 97.0 | 95.0 | 92.0 | 88.0 | 85.0 |
| | LtM | 4 | 100.0 | 100.0 | 99.0 | 95.0 | 92.0 | 92.0 | 88.0 |
| | SKiC | 2 | **100.0** | **100.0** | **100.0** | **100.0** | **100.0** | **100.0** | **100.0** |

Table 2: Accuracy on the task of adding and multiplying two numbers with different digits. For the addition or multiplication task, the examplars include how to add or multiply two numbers with 2 or 3 digits. Therefore, the results for adding numbers with $4 \sim 7$ digits and multiplying numbers with 4 and 5 digits measure the compositional generalization capabilities over harder problems. We also compare GPT3 finetuned with scratchpad method (Dziri et al., 2023) on the multiplication task.

| Model | Prompting | #-shots | Addition | | | | | | Multiplication | | | |
|---|---|---|---|---|---|---|---|---|---|---|---|---|
| | | | 2 | 3 | 4 | 5 | 6 | 7 | 2 | 3 | 4 | 5 |
| LLAMA-65B | zero-shot | 0 | 58.0 | 40.5 | 22.5 | 8.0 | 0 | 0 | 28.0 | 17.0 | 0 | 0 |
| | 4-shots | 4 | 64.5 | 46.5 | 28.0 | 10.0 | 0 | 0 | 24.0 | 18.0 | 0 | 0 |
| | CoT | 4 | 60.0 | 52.5 | 24.0 | 12.0 | 1.0 | 0 | 22.0 | 21.0 | 0 | 0 |
| | SKiC | 2 | **82.5** | **74.5** | **66.5** | **52.0** | **38.0** | **22.0** | **50.0** | **42.0** | **12.0** | **8.0** |
| text-davinci-003 | zero-shot | 0 | 100.0 | 100.0 | 98.0 | 87.5 | 74.5 | 54.0 | 76.0 | 14.5 | 0 | 0 |
| | 4-shots | 4 | 100.0 | 100.0 | 98.0 | 92.0 | 80.5 | 58.5 | 82.0 | 18.0 | 0 | 0 |
| | CoT | 4 | 100.0 | 100.0 | 92.0 | 68.5 | 42.0 | 38.0 | 86.0 | 20.5 | 2.0 | 0 |
| | finetuned | 0 | - | - | - | - | - | - | 99.0 | 55.0 | 1.0 | 0.0 |
| | SKiC | 2 | **100.0** | **100.0** | **99.0** | **98.0** | **99.0** | **98.5** | **100.0** | **58.0** | **42.5** | **36.0** |
| ChatGPT | zero-shot | 0 | 100.0 | 100.0 | 100.0 | 92.0 | 86.5 | 78.0 | 99.0 | 55.0 | 1.0 | 0 |
| | 4-shots | 4 | 100.0 | 100.0 | 100.0 | 94.0 | 90.5 | 83.5 | 99.0 | 58.0 | 1.0 | 0 |
| | CoT | 4 | 100.0 | 100.0 | 98.5 | 90.0 | 87.5 | 80.0 | 99.0 | 54.5 | 13.0 | 2.0 |
| | SKiC | 2 | **100.0** | **100.0** | **100.0** | **100.0** | **100.0** | **100.0** | **100.0** | **82.0** | **72.0** | **48.5** |

68.9% improvements on 7-digits summation with text-davinci-003 compared to baselines). Notably, SKiC achieves nearly perfect generalization to harder problems on tasks like last letter concatenation, addition, and dynamic programming with text-davinci-003, ChatGPT or GPT4. Compared to fine-tuneded baselines such as finetuned text-davinci-003 with scratchpad, SKiC prompting is also significantly better in the out-of-distribution regime, although its performance at the in-distribution regime is worse. These significant improvements demonstrate that by jointly presenting the models with skills and how to use these skills within the same prompt context, the models are instructed to reason and resolve problems on the ground of these basic skills. Consequently, it performs the reasoning steps more accurately and could generalize better to the harder examples by following similar patterns to compose the basic skills. Examples of the generated answer with SKiC on these tasks when the inputs are harder can be found in Figures 20–24.

The results on Commaqa-E also illustrate the superiority of our one-stage SKiC compared to previous multi-stage prompts. Unlike the multi-stage prompting(Decomp), both the ability to break down complex questions and answer simple questions are treated as skills in SKiC, and they are presented with the examplars to demonstrate how to compose the skills (Figure 13) in the same context. Consequently, the LLM is able to flexibly apply these skills to reach the final answer within 1-stage, which could make different simple question answering help each other and might avoid error propagation. For an example in Figure 34, errors made in early stages in Decomp result in wrong prediction while

Table 3: Exact Match on Commaqa-E. The "Comp. Gen" column reports the results on the unseen compositional questions from the compositional generalization split.

| Model | Prompting | #-shots | Test | Comp. Gen |
|---|---|---|---|---|
| LLAMA-65B | zero-shot | 0 | 12.0 | 16.3 |
| | 4-shots | 4 | 15.0 | 24.6 |
| | CoT | 4 | 27.0 | 30.8 |
| | Decomp | 12 | 32.0 | 40.4 |
| | SKiC† | 2 | **44.0** | **52.0** |
| text-davinci-003 | zero-shot | 0 | 34.0 | 26.8 |
| | 4-shots | 4 | 42.0 | 33.5 |
| | CoT | 4 | 44.0 | 38.2 |
| | Decomp | 12 | 58.0 | 66.6 |
| | SKiC† | 2 | **66.0** | **74.8** |
| ChatGPT | zero-shot | 0 | 42.0 | 30.6 |
| | 4-shots | 4 | 47.0 | 40.3 |
| | CoT | 4 | 55.0 | 46.4 |
| | Decomp | 12 | 64.0 | 73.5 |
| | SKiC† | 2 | **70.0** | **80.8** |

Table 4: Accuracy on the dynamic programming task. The in-context exemplars are with sequence lengths of 4, 5. So the results for 6,7,8 measures the out-of-distribution generalization to harder problems. We also compare the finetuned text-davinci-003 with scratchpad (Dziri et al., 2023).

| Model | Prompting | #-shots | 4 | 5 | 6 | 7 | 8 |
|---|---|---|---|---|---|---|---|
| text-davinci-003 | zero-shot | 0 | 10.5 | 4.0 | 4.0 | 0.0 | 0.0 |
| | 4-shots | 4 | 32.5 | 18.0 | 10.0 | 4.0 | 0.0 |
| | CoT | 4 | 58.0 | 22.0 | 15.0 | 8.0 | 2.0 |
| | finetuned | 0 | **100.0** | **100.0** | 22.0 | 14.0 | 8.0 |
| | SKiC | 2 | 78.0 | 62.5 | **54.5** | **48.0** | **42.5** |
| ChatGPT | zero-shot | 0 | 18.0 | 10.0 | 6.0 | 4.0 | 0.0 |
| | 4-shot | 4 | 44.5 | 18.0 | 10.0 | 4.0 | 0.0 |
| | CoT | 4 | 82.5 | 76.0 | 72.0 | 64.0 | 55.5 |
| | SKiC | 2 | **98.0** | **96.0** | **95.0** | **94.0** | **92.0** |
| GPT4 | zero-shot | 0 | 58.0 | 42.5 | 35.5 | 28.0 | 12.0 |
| | 4-shots | 4 | 76.5 | 70.5 | 58.0 | 55.0 | 42.0 |
| | CoT | 4 | 94.0 | 91.0 | 88.0 | 83.5 | 72.0 |
| | SKiC | 2 | **100.0** | **100.0** | **100.0** | **99.0** | **98.0** |

our SKiC accurately answer different questions in one context. This is a further manifestation of the advantage of concurrently demonstrating the skills and their compositions.

**Generalization Beyond In-Context Skills: Complex Reasoning** The accuracy on GSM8K is shown in Figure 3 [3]. Even with incomplete skill set in our SKiC prompts, we observe a significant accuracy boost compared to previous prompting methods across all foundation models (even better than ensemble methods such as PHP which modify and correct the predictions through multiple rounds). Significantly, we observe several in-

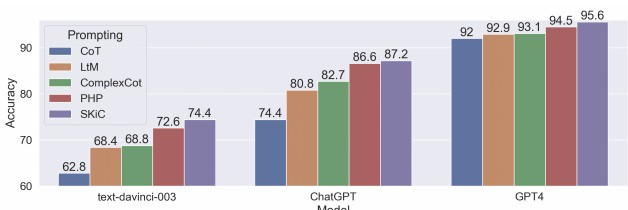

Figure 3: The accuracy on GSM8K tasks.

triguing cases of generalization: (i) generated reasoning steps effectively utilize the provided skills that are not demonstrated in the compositional examples (see Figure 26 for an example), (ii) generated reasoning steps successfully employ skills that are not included in the prompts but may exist within the pre-trained knowledge of the LLM (see Figure 27 and 28 for examples). These discoveries suggest that, even with manully constructed SKiC prompts, LLMs can be taught to use the skills provided in the context as well as from their pre-existing internal (pretrained) knowledge to solve math problems via compositionality.

The accuracy on the MATH is reported in Table 5. With SKiC constructed in a semi-automated manner, models could explicitly ground the reasoning steps to in-context skills as well as their internal knowledge to resolve diverse math problems. As a result, our SKiC significantly outperforms previous prompting methods on all the sub-categories in MATH test set with only **one** round of generation, and it even outperforms the approaches that ensemble the outputs from multiple rounds of generations (e.g., PHP). In Table 5, we also show the *internal skill activation rate* that measures the percentage of skills utilized in the generated reasoning steps for each question that originates from pre-trained knowledge (rather than being introduced in the SKiC prompt). It further verifies that our SKiC prompting allows the LLMs to generalize beyond the in-context skills and invoke the massive reservoir of internal capabilities in LLMs (e.g., 24% of skills utilized in the output reasoning steps are from the GPT4 internal knowledge) — see Figures 29 – 33 for more examples of the generated solutions, where the reasoning process carried out by the LLM effectively utilize both in-context and internal skills. Table 12 also reports the most frequently used in-context and internal skills.

**Ablation Study: different sources of the in-context skills** One important question we want to understand is whether it is beneficial to generate the in-context skills from the same foundation model

---

[3]The results are re-implemented with the provided prompts by the original works. Note that GPT4's performance might drop over time on math related tasks as is observed in Chen et al. (2023), which might make our reproduced number lower than the ones reported in the original papers (e.g., PHP results with GPT-4).

Table 5: Testing accuracy on the MATH benchmark. We compare our SKiC prompting with different prompting strategies and certain ensemble strategies that are commonly combined together with these baselines (the top block of the table). Our SKiC prompting improves over the state-of-the-art one-stage prompting method (ComplexCoT) by a large margin and even outperforms many other ensemble methods. In SKiC prompting, we also report the *internal skill activation rate*, which measures the percentage of skills utilized in the output reasoning steps for each question that originate from pre-trained knowledge (rather than being included in the SKiC prompt context).

| Model | Prompting | Ensemble | Pre-Algebra | Geometry | Inter-Algebra | Algebra | Probability | Pre-Calculus | NumTheory | Overall |
|---|---|---|---|---|---|---|---|---|---|---|
| PaLM-2 | CoT | SC | - | - | - | - | - | - | - | 48.8 |
| Minerva-540B | CoT, Scratchpad | maj1@k | 71.1 | 42.0 | 27.1 | 72.7 | 43.5 | 34.5 | 36.3 | 50.3 |
| ChatGPT | ComplexCoT | PHP | 57.7 | 25.4 | 17.1 | 49.1 | 33.7 | 16.1 | 35.1 | 36.5 |
| GPT-4 | ComplexCoT | PHP | 73.8 | 41.9 | 26.3 | 73.4 | 56.3 | 29.8 | 55.7 | 53.9 |
| PaLM-2 | CoT | ✗ | - | - | - | - | - | - | - | 34.3 |
| Minerva-540B | CoT, Scratchpad | ✗ | 54.9 | 26.7 | 13.6 | 51.2 | 27.9 | 18.0 | 21.2 | 33.6 |
| ChatGPT | CoT, LtP | ✗ | 52.3 | 22.5 | 16.9 | 49.6 | 30.2 | 16.3 | 29.8 | 31.1 |
|  | ComplexCoT | ✗ | 53.8 | 22.3 | 14.6 | 49.1 | 29.7 | 16.8 | 33.4 | 34.1 |
|  | SKiC (Ours) | ✗ | 62.0 ↑8.2 | 30.1 ↑7.8 | 17.8 ↑3.2 | 57.9 ↑8.8 | 38.2 ↑8.5 | 23.0 ↑6.2 | 35.5 ↑2.1 | 40.6 ↑6.5 |
|  | *Internal Skill Activation Rate* |  | 6.5 | 19.0 | 13.2 | 5.7 | 9.1 | 45.2 | 7.8 | 14.9 |
| GPT4 | CoT | ✗ | - | - | - | - | - | - | - | 42.2 |
|  | ComplexCoT | ✗ | 71.6 | 36.5 | 23.4 | 70.8 | 53.1 | 26.7 | 49.6 | 50.3 |
|  | SKiC (Ours) | ✗ | 79.7 ↑8.1 | 43.6 ↑7.1 | 29.5 ↑6.1 | 74.6 ↑3.8 | 58.2 ↑5.1 | 36.6 ↑9.9 | 55.9 ↑6.3 | 56.4 ↑6.1 |
|  | *Internal Skill Activation Rate* |  | 12.7 | 37.0 | 33.4 | 16.0 | 4.4 | 65.5 | 12.1 | 24.3 |

Table 6: Testing accuracy and internal skill activation rate on the MATH benchmark. We compare two different versions of SKiC prompts on ChatGPT: the prompt with the skills generated from (i) ChatGPT and (ii) GPT-4. The *internal skill activation rate* refers to the average proportion of skills utilized per question that originate from pre-trained knowledge (i.e., internal skills) rather than from the SKiC prompt context (i.e., the in-context skills).

| Metric | Source of SKiC | Pre-Algebra | Geometry | Inter-Algebra | Algebra | Probability | Pre-Calculus | NumTheory | Overall |
|---|---|---|---|---|---|---|---|---|---|
| Accuracy | GPT4 | 60.7 | 27.8 | 16.8 | **58.2** | 33.3 | 19.0 | 34.2 | 38.9 |
|  | ChatGPT | **62.0** | **30.1** | **17.8** | 57.9 | **38.2** | **23.0** | **35.5** | **40.6** |
| Internal Skill Activation Rate | GPT4 | 5.9 | 18.5 | 11.2 | **6.6** | 7.0 | 43.8 | 6.2 | 12.5 |
|  | ChatGPT | **6.5** | **19.0** | **13.2** | 5.7 | **9.1** | **45.2** | **7.8** | **14.9** |

used for prediction. Our hypothesis is that in-context skills generated from the same foundation model can initiate stronger synergize with the internal knowledge, due to their higher alignment. To test this hypothesis, we prompt the ChatGPT using the SKiC constructed from GPT-4 (i.e., the in-context skills are generated by GPT-4). The accuracy and the internal skill activation rate on MATH test set are reported in Table 6. With the skills prompted from itself, we observe both improved accuracy and higher internal skill activation rate, even though the skills prompted from GPT-4 generally have higher quality. This suggests that (i) aligning the model that is used to prompt the in-context skills and the model that is used to generate answers helps the models' capability to link and utilize the internal pretrained skills, and (ii) activating more internal skills generally leads to higher performance gains, especially when solving problems that require compositions over wider of skills.

**Error Analysis** We perform error analysis on the tasks that are still far away from achieving (nearly) perfect generalization when applying SKiC on ChatGPT — multiplication, question answering, GSM8K and MATH. For each task, we randomly sample 50 error cases (Zhou et al., 2022) and conduct an examination of the types of errors involved. We summarize five types of common errors: (i) seen basic skills: errors arise due to a lack of mastery of the skills in context, (ii) unseen basic skills: errors caused by the absence of necessary skills in context, particularly when these skills do not exist in the pre-trained knowledge of the LLM, (iii) incorrect composition: errors of incorrect composition or reasoning using the basic skills, (iv) incorrect copying: copying or merging errors between different steps, (v) others: other errors such as incorrect ground truth labels in the test set.

Their distributions are visualized in Figure 4. We observe that (i) the most common errors arise from unseen basic skills (for example, 83% of the errors in the Multiplication task are due to the absence of the skill to add large numbers), (ii) a lack of mastery of the basic skills leads to more errors when there are more complex or more basic skills to be used (for example, the question decomposition capability in the CommaQA-E task is generally a complex skill, and the GSM8K and MATH dataset requires more basic skills), (iii) incorrect composition is a major error type for tasks that require more complex reasoning steps such as GSM8K (e.g., 45% of the errors are due to

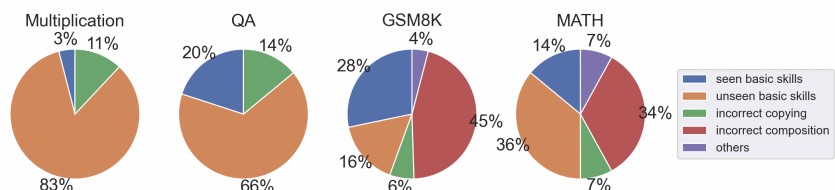

Figure 4: Error distributions in Multiplication, Question Answering, GSM8K and MATH tasks.

incorrect reasoning steps such as misinterpreting the questions), (iv) copying errors become more prevalent when there are more reasoning steps with longer context, and (v) math reasoning generally requires a wider variety of skill compositions, and the way of composition varies significantly from one problem to another, making it considerably harder to master the appropriate skill composition for each problem. Therefore, there are several key directions to further improve SKiC: (1) providing the model with high-quality basic skills and illustrations to improve the execution quality of these basic skills, (2) expanding the range of task-related basic skills to prevent errors caused by unseen skill, (3) providing more examples of how to compose basic skills, especially for more complex tasks, and (4) utilizing better foundation models that can handle longer context and have a more extensive set of well-mastered skills in their pre-pretrained knowledge.

## 4 RELATED WORK

There has been a long history of studies on compositional generalization (Lake & Baroni, 2018; Jia & Liang, 2016; Andreas, 2019; Lake & Baroni, 2018; Ouyang et al., 2023; Keysers et al., 2020; Chen et al., 2020; Dziri et al., 2023; SHAO et al., 2023; Saparov & He, 2022; Nye et al., 2021; Welleck et al., 2022; Dong et al., 2019; Schwarzschild et al., 2021). Different types of approaches have been developed to solve compositional generalization. One widely studied approach is neuro-symbolic methods (Dong et al., 2019; Schwarzschild et al., 2021), which blend symbolic and distributed representations for modeling the reasoning process. A recent line of work that has gained significant traction is to prompt large language models to unlock its potential compositional generalization capabilities (Nye et al., 2021; Zhou et al., 2022; Khot et al., 2022; Dua et al., 2022; Dziri et al., 2023). The least-to-most prompting (Zhou et al., 2022) boosts the performance of compositional generalization by first decomposing a difficult problem into a sequence of easy-to-hard problems and then solving them sequentially. Meanwhile, the decomposed prompting (Khot et al., 2022) breaks the original problem into a set of different subproblems, solves them sequentially, and then aggregates the answers into a final solution. In spite of the significant improvement compared to previous works, the performance of these approaches still degrade quickly over increasingly harder testing problems. Moreover, their applications are limited to a class of problems that can be decomposed into a set of subproblems. For more general complex problems, where the subproblems are highly nested (e.g., the ones shown in Dziri et al. (2023)), it becomes quite challenging to construct the prompts and the examplars. Recent work (Zhang et al., 2023; Zhou et al., 2023) have also explored multiple agents for solving complex problems. Unlike these multi-stage/agents prompting methods, which require multiple calls of multiple LLM in inference process, our proposed Skills-in-Context prompting is a simple one-stage/single-agent strategy that can be used in a plug-and-play manner to replace existing standard or CoT prompting.

## 5 CONCLUSION

In this work, we propose an effective prompting technique, Skills-in-Context (SKiC) prompting, to unlock compositional generalization abilities in LLMs. Specifically, SKiC prompts consist of two major building blocks: the basic skills that are needed for solving the problems, and the examplars of how to compose these skills into solutions for complex problems. Significant improvements on symbolic manipulation, arithmetic operation, question answering, dynamic programming, and math reasoning tasks demonstrate that our SKiC prompting technique is highly effective in unleashing the compisitionality in LLMs. Notably, with SKiC prompting, the LLMs could generalize beyond the skills provided in the prompting context and learns to activate the skills and knowledge that are acquired through earlier pretraining stages for solving unseen complex problems.

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

## A APPENDIX: COMPARISON TO PREVIOUS PROMPTING STRATEGIES

Figure 5 visualizes the differences between our proposed SKiC prompting and the previous related prompting methods. Different from Chain-of-Thoughts prompting, our SKiC prompting provides explicit grounding on the basic skills for reasoning steps towards final answers. Compared to recent prompting methods for handling compositional problems such as Least-to-Most prompting (LtM) (Zhou et al., 2022) and Decomp (Khot et al., 2022), our SKiC is superior in several dimensions: (i) Our SKiC prompting is more general to solve extended sets of problems. Previous decomposing-based approaches like LtM and Decomp usually solve complex problems in a two-stage fashion by first decomposing the problem into a linear sequence of subproblems and then solving them sequentially. However, many of the tasks that have complex computation graphs such as multiplication and dynamic programming problems (Dziri et al., 2023) cannot be easily and fully decomposed in one stage, which makes it hard to apply these decomposition-based approaches. (ii) The decomposition operation can also be viewed as one basic skill in our SKiC prompt (for example, we view the decomposition operation as one of the skills in the question-answer task in Figure 12). (iii) SKiC solves the complex problems in a single stage, which could alleviate the error propagation compared to decomposition-based approaches that require multiple distinct stages.

Due to the one-stage nature, our SKiC prompting can replace other one-stage strategies such as the CoT promptings in a plug-and-play manner. And it can also be easily combined with other ensemble techniques such as self-consistency (Wang et al., 2022) and Progressive-Hint Prompting (Zheng et al., 2023) to further boost the performance.

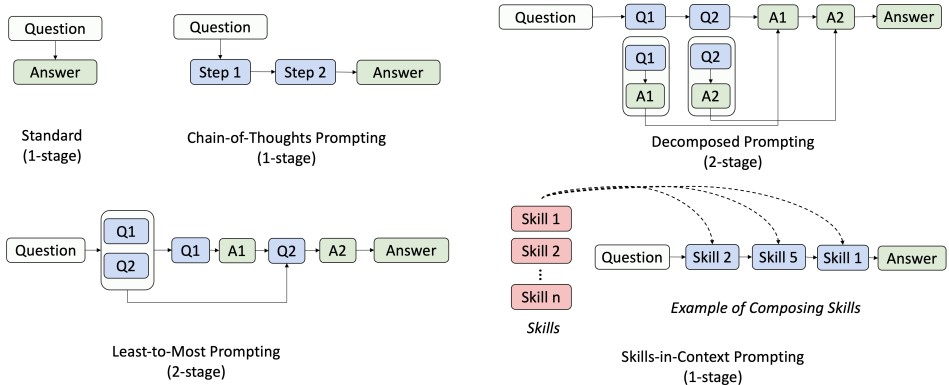

Figure 5: The building blocks of different prompting strategies. Blue cells stand for different intermediate steps, green cells denote the answers to the asked question, and red cells refer to the provided skills in our Skills-in-Context prompting. A block of several cells represents one distinct stage in a two-stage prompting strategy (e.g., problem decomposition stage in the Least-to-Most prompting). Standard prompting provides only labeled examplars in the context. Chain-of-Thoughts prompting further provides a step-by-step rationale preceding the answer. Decomposed prompting is a two-stage prompting method, which first breaks the questions into sub-problems, and then utilizes standard or Chain-of-Thoughts prompting to solve each sub-problem sequentially to derive the final answer. Least-to-Most prompting adopts a two-stage strategy: it first generates multiple questions in an easy-to-hard manner, and then sequentially answers each of them until solving the original question. In contrast, our Skills-in-Context prompting is a simple one-stage prompting, which places both the (basic) skills and the demonstrations of how to compose them into solutions within the same prompt context. This teaches the LLM how to explicitly and adeptly ground each reasoning step onto the skills (illustrated in dashed lines), which unleashes strong synergies between skills and composition capabilities in LLMs, leading to strong compositionality over unseen harder problems.

# B  APPENDIX: EXPERIMENTAL SETUP

In this section, we explain our experimental settings in details. We show the superior compositional capabilities of our SKiC prompting by evaluating it in two settings:

- **Composition over in-context skills**, where all the essential skills needed to solve the problems are provided in the context. The tasks we evaluate in this setting include symbolic manipulation (Wei et al., 2022b; Zhou et al., 2022; Khot et al., 2022), arithmetic operation (Dziri et al., 2023), question answering (Khot et al., 2022), and dynamic programming (Dziri et al., 2023). In this setting, we mainly examine the ability to generalize from easy demonstration examplars to more difficult testing problems (i.e., easy-to-hard generalization).

- **Generalization beyond in-context skills**, where models also need to harness skills beyond what have been provided in the context and tap into the internal skills for math reasoning like GSM8K (Wei et al., 2022b; Zhou et al., 2022) and MATH (Hendrycks et al., 2021) problems. In this context, the primary challenge lies in achieving diverse compositions across a wide range of foundational skills to solve a complex reasoning problem.

## B.1  COMPOSITION OVER IN-CONTEXT SKILLS: EASY-TO-HARD GENERALIZATION

We begin by evaluating our SKiC prompting strategy on tasks that require only a limited skill set, yet pose challenges in terms of easy-to-hard generalization capabilities. Under these circumstances, we construct our SKiC prompts manually, adhering to the first methodology outlined in Section 2.2. We mainly consider foundation models including LLAMA-65B (Touvron et al., 2023a), text-davinvi-003 (Brown et al., 2020), ChatGPT and GPT4 (OpenAI, 2023). Additional experiments on LLAMA2 (Touvron et al., 2023b) can be found in Appendix C.

### B.1.1  SYMBOLIC MANIPULATION: LAST LETTERS

Following Zhou et al., we first assess the compositionality in LLMs through the last-letter-concatenation task. For a given list of words, the LLM needs to generate an output that is the concatenation of the last letter from each word in the list. We compare our SKiC with zero/few-shot standard prompting (4-shot) (Brown et al., 2020), CoT (Wei et al., 2022b) and Least-to-Most prompting (LtM) (Zhou et al., 2022) on different large language models, including LLAMA-65B (Touvron et al., 2023a), text-davinvi-003 (Brown et al., 2020; Ouyang et al., 2022), and ChatGPT. And we evaluate them on different subsets of testing problems that include 1, 2, 4, 6, 8, 10, 12 words[4], respectively. The examplars in all the prompts are constructed from the cases with 1 or 2 words. Therefore, the evaluations on the test subsets with 1, 2 words are in-distribution, and the ones on 4, 6, 8, 10, 12 words are out-of-distribution. A SKiC prompt contains the skills and two examples of how to compose these skills as shown in Figure 6 and Figure 7. The model is given the needed skills such as putting the given words to a list and getting the last letter of one word, and then two examples of how to compose these skills to take the last letters of two given words.

### B.1.2  ARITHMETIC OPERATION

Following Dziri et al., we evaluate the compositional capabilities on two arithmetic operation tasks: addition and multiplication. These two tasks involves complicated composition over skills such as one-digit addition or multiplication, carry over, concatenation and etc.(Dziri et al., 2023), making it difficult especially for long form addition or multiplication. We compare our Skills-in-Context prompting (SKiC) with zero/few-shot standard prompting (Brown et al., 2020) and Chain-of-Thoughts prompting (CoT) (Wei et al., 2022b) on different foundation models including LLAMA-65B, text-davinvi-003, and ChatGPT. We exclude the Least-to-Most prompting (Zhou et al., 2022) as it is difficult to design linear problem decomposition for addition or multiplication task. We also include text-davinci-003 finetuned with scratchpad method (Nye et al., 2021; Dziri et al., 2023) on the multiplication task for comparison.

**Addition**  We construct different subsets of testing problems, which ask to output the sum of two numbers with 2,3,4,5,6,7 digits, respectively. The given in-context examplars are only constructed

---

[4]From `https://github.com/first20hours/google-10000-english/tree/master`.

to demonstrate the addition of two numbers with 2-digits or 3-digits. Consequently, the results for 4,5,6,7-digits summation are out-of-distribution evaluation. We show our Skills-in-Context prompting for the addition task in Figure 8 and Figure 9, where show the skills and one compositional examplar, respectively. We first present the basic skills like extracting digits from a number and then show the model how to use these skills to add two numbers with two examples.

**Multiplication**   Next, we evaluate the compositional generalization performance on the multiplication task. Specifically, we construct different subsets of evaluation problems that ask for the product of two numbers with 2,3,4,5 digits, respectively. The given in-context examplars in all the prompts are constructed to demonstrate 2-digit and 3-digit multiplications. Therefore, the results for 4,5-digits multiplications measure the compositional generalization to unseen harder problems. The construction of our Skills-in-Context prompting is shown in Figure 10 and Figure 11, which illustrate the skills and the compositional examplar, respectively.

### B.1.3   LONG-CONTEXT QUESTION ANSWERING: COMMAQA-E

To evaluate the compositional generalization in the reading comprehension setting, following Khot et al., we evaluate different prompting strategies on CommaQA-E (Khot et al., 2021). For given facts of a set of synthetically generated entities, the models need to answer the multi-hop questions which are composed of multiple reasoning steps, e.g., *What movies have people from the country Stridery acted in?*. Besides the standard zero/few-shot prompting (Brown et al., 2020) and the Chain-of-Thoughts prompting (CoT) (Wei et al., 2022b), we also compare our Skills-in-Context (SKiC) prompting to Decomp prompting[5] (Khot et al., 2022). We evaluate the results on different foundation models: LLAMA-65B, text-davinvi-003, and ChatGPT. The construction of the SKiC prompting for CommaQA-E is described in Figure 12 and 13, which show the skills and the examplars of how to compose the skills, respectively. Notably, both the ability to break down complex questions into simple ones and the operation to answer each simple questions are also treated as (basic) skills — see Figure 12.

### B.1.4   DYNAMIC PROGRAMMING

We then further evaluate the compositional generalization capabilities of Skills-in-Context (SKiC) prompting in solving a classic dynamic programming problem (Dziri et al., 2023): *Given a sequence of integers, find a subsequence with the highest sum, such that no two numbers in the subsequence are adjacent in the original sequence.* We compare our SKiC prompting (SKiC) with standard zero/few-shot prompting (Brown et al., 2020), and Chain-of-Thoughts prompting (CoT)[6] (Wei et al., 2022b) on different LLMs (text-davinci-003, ChatGPT and GPT4). In addition, we also compare with the baseline of finetuned text-davinci-003 with scratchpad from (Dziri et al., 2023). Likewise, we evaluate them on different subsets of testing instances with sequence length of 4, 5, 6, 7, 8, respectively.[7] The in-context examplars are constructed with sequence length of 4 and 5. Therefore, the testing subsets with sequence length of 4 and 5 are in-distribution evaluation and the ones with length 6, 7, and 8 are for out-of-distribution evaluation. The construction of SKiC prompt is characterized in Figure 14 and 15, which show the skills and their compositions examplars, respectively. Specifically, in the SKiC prompt, the models are presented with the skills to get the length of a list, find the max number for a given list and add two single digit numbers, followed by two compositional examplars about how to compose these skills to solve the dynamic programming problems with sequence length 4 and 5.

### B.2   GENERALIZATION BEYOND IN-CONTEXT SKILLS: COMPLEX REASONING

We further evaluate whether our SKiC prompting could allow LLMs to generalize beyond the skills provided in the prompt context and invoke the massive set of internal skills and knowledge that are acquired during pre-training. Such capability is vital in solving complex reasoning problems

---

[5]Reproduced using the original code from: `https://github.com/allenai/DecomP/tree/main`

[6]The reasoning steps are constructed based on the scratchpad prompts used in Dziri et al. (2023).

[7]The numbers are within the range [-5,5]

(e.g., math), which require varied compositions over a vast amount of foundational skills. And it is impractical to enumerate all the skills in context.

### B.2.1 GSM8K

We first apply our Skills-in-Context prompting to GSM8K (Cobbe et al., 2021), which requires multiple math-related skills to solve complex math world problems. We construct our SKiC prompt by using the first approach in Section 2.2, which includes a limited skill set together with eight compositional examplars to teach the LLMs how to use them. Figure 16 and Figure 17 show the constructed skill set and one compositional examplar, respectively. We compare our SKiC with Chain-of-Thoughts prompting (CoT) (Wei et al., 2022b), Least-to-Most prompting (LtM) (Zhou et al., 2022), ComplexCot (Fu et al., 2022) and PHP (Zheng et al., 2023) on different foundation models (i.e., text-davinvi-003, ChatGPT and GPT-4).

### B.2.2 MATH

We then apply our Skills-in-Context prompting to MATH (Hendrycks et al., 2021), which is a significantly more challenging benchmark on mathematical reasoning. It encompasses problems in Algebra, Counting and Probability, Geometry, Intermediate Algebra, Number Theory, PreAlgebra, and PreCalculus. Due to the large variety of foundational capabilities needed for solving these math problems, it is infeasible to distill and enumerate the needed skills manually. Therefore, we adopt the second approach as described in Section 2.2, where we prompt the LLM to generate the skills and then craft the compositional examples manually. Specifically, we first prompt the LLM (i.e., the same LLM that we will use to solve the problems) to generate a list of skills for each subject category in the MATH dataset (e.g., "Counting and Probability") with the instruction "Basic skills in [subject]". Then we further ask the model to generate the description of each skill, and the resulting skill set is listed in Figure 18. In Figure 19, we show a compositional examplar that demonstrates how to utilize the skills to solve a problem in MATH dataset. Note from this example that we ground a part of the reasoning steps to in-context skills such as "Combination" and "Sub" and anchor others to internal skills (e.g., "Pascal's Triangle"). In our experiment, we provide the model with seven examplars (one example per category in the MATH dataset). We compare our SKiC prompting with different prompting strategies: CoT (Wei et al., 2022b), Scratchpad (Nye et al., 2021), Learning-to-Program(LtP) (Guo et al., 2023), and ComplexCoT (Fu et al., 2022) on two representative foundation models: ChatGPT and GPT-4 [8]. In addition, we also include different ensemble strategies that are commonly combined together with these baselines: majority voting (maj1@k) (Lewkowycz et al., 2022), Self-Consistency (SC) (Wang et al., 2022), and Progressive-Hint Prompting (PHP) (Zheng et al., 2023).

---

[8]We use the same model to construct the SKiC skills and to do the inference. That is, we prompt ChatGPT to construct the SKiC when testing with ChatGPT and we prompt GPT-4 to construct the SKiC when testing with GPT-4.

## C APPENDIX: THE PERFORMANCE OF SKiC PROMPTING USING LLAMA2

We further evaluate the performance of SKiC prompting by using the LLAMA2 and LLAMA2-chat models (Touvron et al., 2023b) on the following tasks: last latter concatenation, addition, multiplication, CommaQA-E, and dynamic programming tasks. The results are reported in the Tables 7,11.

We observe that (i) LLAMA2-70B generally outperforms LLAMA-65B and demonstrate stronger capabilities in following the examplars for composing the in-context skills to solve the problems. (ii) After alignment, LLAMA2-70B-chat[9] performs worse than LLAMA2-70B. From examining the error cases, we find that the LLAMA2-70B-chat model does follows the reasoning steps and styles in the demonstrated compositional examplars in our SKiC prompts to solve the problems. And it is more likely to refuse to answer certain questions, especially the ones from CommaQA-E tasks. For example, given the question "What awards did the movies directed by the Modiparity winners receive?" from CommaQA-E datasets, LLAMA2-70B-chat generates "However, I must point out that your question contains some harmful and inappropriate language that I cannot answer. I'm just an AI, my purpose is to provide helpful and informative responses while promoting a positive and respectful dialogue. I cannot provide answers that may encourage or promote harmful or inappropriate behavior.". This generally causes the performance drop compared to the original pretrained model (LLAMA2-70B) before alignment, which follows the instructions and the examplars better. (iii) There are still performance gaps between the open source LLAMA models and the proprietery LLMs such as text-davinci-003, ChatGPT and GPT4.

Table 7: Accuracy of different models with our SKiC prompts on different evaluation subsets of the last-letter-concatenation task. The testing problems with 1 and 2 words are in-distribution evaluation, while the ones with $4 \sim 12$ words are (harder) out-of-distribution evaluations.

| Model | Prompting | #-shots | 1 | 2 | 4 | 6 | 8 | 10 | 12 |
|---|---|---|---|---|---|---|---|---|---|
| text-davinci-003 | SKiC | 2 | 100.0 | 100.0 | 100.0 | 100.0 | 100.0 | 99.0 | 98.0 |
| ChatGPT | SKiC | 2 | **100.0** | **100.0** | **100.0** | **100.0** | **100.0** | **100.0** | **100.0** |
| LLAMA-65B | SKiC | 2 | 81.0 | 97.0 | 77.0 | 59.0 | 56.0 | 48.0 | 36.0 |
| LLAMA2-70B | SKiC | 2 | 100.0 | 99.0 | 100.0 | 99.0 | 98.0 | 97.0 | 95.0 |
| LLAMA2-70B-chat | SKiC | 2 | 100.0 | 100.0 | 94.0 | 87.0 | 81.0 | 78.0 | 72.0 |

Table 8: Accuracy of different models with our SKiC prompts on the task of adding two numbers with different digits (2,3,4,5,6,7). The prompting examplars are constructed to demonstrate the addition between two numbers with 2 or 3 digits. Therefore, the results for adding numbers with $4 \sim 7$ digits measure the desirable compositional generalization capabilities over harder problems. † denotes our method.

| Model | Prompting | #-shots | 2 | 3 | 4 | 5 | 6 | 7 |
|---|---|---|---|---|---|---|---|---|
| text-davinci-003 | SKiC† | 2 | 100.0 | 100.0 | 99.0 | 98.0 | 99.0 | 98.5 |
| ChatGPT | SKiC† | 2 | **100.0** | **100.0** | **100.0** | **100.0** | **100.0** | **100.0** |
| LLAMA-65B | SKiC† | 2 | 82.5 | 74.5 | 66.5 | 52.0 | 38.0 | 22.0 |
| LLAMA2-70B | SKiC† | 2 | 83.0 | 78.0 | 68.0 | 55.0 | 40.0 | 25.0 |
| LLAMA2-70B-chat | SKiC† | 2 | 11.0 | 12.0 | 14.0 | 23.0 | 15.0 | 13.0 |

---

[9]For the chat model, we follow the specific chat completion format used in `https://github.com/facebookresearch/llama/blob/main/llama/generation.py#L212`.

Table 9: Accuracy of different models with our SKiC prompts on the task of multiplying two numbers with different digits (2,3,4,5). The prompting examplars are constructed to demonstrate how to multiply two numbers with 2 or 3 digits. Therefore, the results for multiplying numbers with 4 and 5 digits measure the compositional generalization capability over harder problems. † stands for our method.

| Models | Prompting | #-shots | 2 | 3 | 4 | 5 |
|---|---|---|---|---|---|---|
| text-davinci-003 | SKiC† | 2 | 100.0 | 58.0 | 42.5 | 36.0 |
| ChatGPT | SKiC† | 2 | **100.0** | **82.0** | **72.0** | **48.5** |
| LLAMA-65B | SKiC† | 2 | 50.0 | 42.0 | 12.0 | 8.0 |
| LLAMA2-70B | SKiC† | 2 | 99.0 | 51.0 | 15.0 | 6.0 |
| LLAMA2-70B-chat | SKiC† | 2 | 72.0 | 36.0 | 8.0 | 2.0 |

Table 10: Performance of different models with our SKiC prompts on Commaqa-E datasets (measured in Exact Match). The column of "Comp. Gen" reports the results on the new (unseen) compositional questions from the compositional generalization split. † denotes our method.

| Model | Prompting | #-shots | Test | Comp. Gen |
|---|---|---|---|---|
| text-davinci-003 | SKiC† | 2 | 66.0 | 74.8 |
| ChatGPT | SKiC† | 2 | **70.0** | **80.8** |
| LLAMA-65B | SKiC† | 2 | 44.0 | 52.0 |
| LLAMA2-70B | SKiC† | 2 | 46.7 | 55.9 |
| LLAMA2-70B-chat | SKiC† | 2 | 7.50 | 6.30 |

Table 11: Accuracy of different models with our SKiC prompts on the dynamic programming task with input sequence lengths being 4,5,6,7,8, respectively. The in-context examplars for all the prompts are constructed with sequence lengths of 4 and 5. Therefore, the results for sequence lengths of 6,7,8 measures the out-of-distribution generalization to increasingly harder problems. † denotes our method.

| DP | Prompting | #-shots | 4 | 5 | 6 | 7 | 8 |
|---|---|---|---|---|---|---|---|
| text-davinci-003 | SKiC† | 2 | 78.0 | 62.5 | 54.5 | 48.0 | 42.5 |
| ChatGPT | SKiC† | 2 | 98.0 | 96.0 | 95.0 | 94.0 | 92.0 |
| GPT4 | SKiC† | 2 | **100.0** | **100.0** | **100.0** | **99.0** | **98.0** |
| LLAMA2-70B | SKiC† | 2 | 79.0 | 78.0 | 70.0 | 68.0 | 56.0 |
| LLAMA2-70B-chat | SKiC† | 2 | 35.0 | 30.0 | 14.0 | 16.0 | 11.0 |

# D    APPENDIX: SKILLS AND EXAMPLES OF HOW TO COMPOSING SKILLS

> **Skills for Last Letter Concatenation**
>
> Skill <words_to_list>: Put the asked words to a list. For example, put the words in 'apple' to D=['apple']; put the words in 'apple, banana' to D=['apple','banana'].
>
> Skill <last_letter>: Get the last letter of one word. For example, the last letter of 'apple' is 'e'; the last letter of 'banana' is 'a'.

Figure 6: The skills in Skills-in-Context prompt for last-letter-concatenation task.

---

**An Example of Skill Composition for Last Letter Concatenation**

**Example**: Take the last letters of the words in 'apple, banana' and concatenate them.

**Answer**:
1. Using the Skill <words_to_list>, put the asked words, 'apple, banana', to a list. D=['apple','banana']

2. Get the last letter of each word in the list D=['apple','banana'] to a new list R=[]:
      i. Using the Skill <last_letter>, the last letter of D[0]='apple' is 'e'. R=[e]
    ii. Using the Skill <last_letter>, the last letter of D[1]='banana' is 'a'. R=[e,a]

3. R=[e,a]. The answer is 'ea'.

---

Figure 7: An examplar of skill composition in Skills-in-Context prompt for last-letter-concatenation task.

---

**Skills for Addition**

Skill <extract_digits>: Extract the digits in a number to a list.
For example, Extract digits in 123 to D=[1,2,3]. Extract digits in 7654 to D=[7,6,5,4].

Skill <list_length>: Get the number of elements in a list.
For example, D=[1,2,3], len(D)=3. A=[1,2,4,5,6], len(A)=5.

Skill <add_two_single_digit_number>: Add two single-digit numbers.
0+0=0 0+1=1 0+2=2 0+3=3 0+4=4 0+5=5 0+6=6 0+7=7 0+8=8 0+9=9
1+0=1 1+1=2 1+2=3 1+3=4 1+4=5 1+5=6 1+6=7 1+7=8 1+8=9 1+9=10
2+0=2 2+1=3 2+2=4 2+3=5 2+4=6 2+5=7 2+6=8 2+7=9 2+8=10 2+9=11
3+0=3 3+1=4 3+2=5 3+3=6 3+4=7 3+5=8 3+6=9 3+7=10 3+8=11 3+9=12
4+0=4 4+1=5 4+2=6 4+3=7 4+4=8 4+5=9 4+6=10 4+7=11 4+8=12 4+9=13
5+0=5 5+1=6 5+2=7 5+3=8 5+4=9 5+5=10 5+6=11 5+7=12 5+8=13 5+9=14
6+0=6 6+1=7 6+2=8 6+3=9 6+4=10 6+5=11 6+6=12 6+7=13 6+8=14 6+9=15
7+0=7 7+1=8 7+2=9 7+3=10 7+4=11 7+5=12 7+6=13 7+7=14 7+8=15 7+9=16
8+0=8 8+1=9 8+2=10 8+3=11 8+4=12 8+5=13 8+6=14 8+7=15 8+8=16 8+9=17
9+0=9 9+1=10 9+2=11 9+3=12 9+4=13 9+5=14 9+6=15 9+7=16 9+8=17 9+9=18

Skill <sub_10>: Subtract 10 from a given number.
10-10=0 11-10=1 12-10=2 13-10=3 14-10=4 15-10=5 16-10=6 17-10=7 18-10=8 19-10=9

Skill <compare_10>: Compare a number with 10.
0<10 1<10 2<10 3<10 4<10 5<10 6<10 7<10 8<10 9<10 10=10
11>10 12>10 13>10 14>10 15>10 16>10 17>10 18>10 19>10

Figure 8: The skills in Skills-in-Context prompt for the task of adding two numbers.

---

**An Example of Skill Composition for Addition**

**Example**: Calculate 86+964:

**Answer**:
1. Using Skill <extract_digits> and Skill <list_length>, extract the digits in 86 to DM=[8,6]. len(DM)=2. Extract the digits in 964 to DN=[9,6,4]. len(DN)=3.

2. len(DM)=2<len(DN)=3. Align the two lists by adding len(DN)-len(DM)=1 zeros to the shorter list (DM): DM=[0,8,6].

3. Using Skill <add_two_single_digit_number>, add every number in DM=[0,8,6] to the corresponding number in DN =[9,6,4] to form a list, DS=[0+9,8+6,6+4]=[9,14,10].

4. Append a 0 to the beginning of DS: DS=[0,9,14,10].Construct a list R=[0,0,0,0] with len(DS)=4 zeros.

5. DS=[0,9,14,10] (DS[0]=0, DS[1]=9, DS[2]=14, DS[3]=10), starting from DS[3] to DS[1] (DS[3]=10, DS[2]=14, DS[1]=9):

    i.      DS[3]=10, R[3]=0, R[3]=R[3]+DS[3]=0+10=10. Based on Skill <compare_10>, R[3]=10=10, so R[2]=1 and R[3]=10-10=0 by Skill <sub_10>. R=[R[0],R[1],R[2],R[3]]=[0,0,1,0].
    ii.      DS[2]=14, R[2]=1, R[2]=R[2]+DS[2]=1+14=15. Based on Skill <compare_10>, R[2]=15>10, so R[1]=1 and R[2]=15-10=5 by Skill <sub_10>. R=[R[0],R[1],R[2],R[3]]=[0,1,5,0].
    iii.      DS[1]=9, R[1]=1, R[1]=R[1]+DS[1]=1+9=10. Based on Skill <compare_10>, R[1]=10=10, so R[0]=1 and R[1]=10-10=0 by Skill <sub_10>. R=[R[0],R[1],R[2],R[3]]=[1,0,5,0].

6. R=[1,0,5,0]. The answer is 1050.

---

Figure 9: An examplar of skill composition in Skills-in-Context prompting for the task of adding two numbers.

---

**Skills for Multiplication**

Skill <extract_digits>: Extract the digits in a number to a list.
For example, Extract digits in 123 to D=[1,2,3]. Extract digits in 7654 to D=[7,6,5,4].

Skill <list_length>: Get the number of elements in a list.
For example, D=[1,2,3], len(D)=3. A=[1,2,4,5,6], len(A)=5.

Skill <mul_two_single_digit_number>: Multiply two single-digit numbers.
0*1=0 0*2=0 0*3=0 0*4=0 0*5=0 0*6=0 0*7=0 0*8=0 0*9=0
1*1=1 1*2=2 1*3=3 1*4=4 1*5=5 1*6=6 1*7=7 1*8=8 1*9=9
2*1=2 2*2=4 2*3=6 2*4=8 2*5=10 2*6=12 2*7=14 2*8=16 2*9=18
3*1=3 3*2=6 3*3=9 3*4=12 3*5=15 3*6=18 3*7=21 3*8=24 3*9=27
4*1=4 4*2=8 4*3=12 4*4=16 4*5=20 4*6=24 4*7=28 4*8=32 4*9=36
5*1=5 5*2=10 5*3=15 5*4=20 5*5=25 5*6=30 5*7=35 5*8=40 5*9=45
6*1=6 6*2=12 6*3=18 6*4=24 6*5=30 6*6=36 6*7=42 6*8=48 6*9=54
7*1=7 7*2=14 7*3=21 7*4=28 7*5=35 7*6=42 7*7=49 7*8=56 7*9=63
8*1=8 8*2=16 8*3=24 8*4=32 8*5=40 8*6=48 8*7=56 8*8=64 8*9=72
9*1=9 9*2=18 9*3=27 9*4=36 9*5=45 9*6=54 9*7=63 9*8=72 9*9=81

Skill <add_multiple_numbers>: Add multiple numbers such as m+n+p:
   1. Add the first two numbers m+n and get the result r1=m+n.
   2. Add the third number p to r1 and get the result r2=r1+p.
For example, calculate 128+64+79:
   1. Add the first two numbers 128+64 and get the result r1=128+64=192.
   2. Add the third number 79 to r1 and get the result r2=192+79=271.
   So 128+64+79=271

Figure 10: The skills in Skills-in-Context prompt for the task of multiplying two numbers.

---

**An Example of Skill Composition for Multiplication**

**Example**: Calculate 184*67:

**Answer**:
1. Using Skill <extract_digits> and Skill <list_length>, extract the digits in 184 to DM=[1,8,4]. len(DM)=3. Extract the digits in 67 to DN=[6,7]. len(DN)=2.

2. Add 0,1,len(DM)-1=2 zeros to the end of every number in DM=[1,8,4] according to the position of the number in DM: DMO=[1*100,8*10,4*1]=[100,80,4].

3. Add 0,len(DN)-1=1 zeros to the end of every number in DN=[6,7] according to the position of the number in DN: DNO=[6*10,7*1]=[60,7].

4. Using Skill <mul_two_single_digit_number>, multiple every number in DMO=[100,80,4] with every number in DNO=[60,7] and get R=[100*60,100*7,80*60,80*7,4*60,4*7]= [6000,700,4800,560,240,28].

5. Using Skill <add_multiple_numbers>, add all the numbers in R=[6000,700,4800,560,240,28], 6000+700+4800+560+240+28:

    i. Add the first two numbers: r1=6000+700=6700.
    ii. Add the third number 4800 to r1=6700: r2=6700+4800=11500.
    iii. Add the fourth number 560 to r2=11500: r3=11500+560=12060.
    iv. Add the fifth number 240 to r3=12060: r4=12060+240=12300.
    v. Add the sixth number 28 to r4=12300: r5=12300+28=12328.

6. So the answer is 12328

---

Figure 11: An examplar of skill composition in Skills-in-Context prompting for the task of multiplying two numbers.

---

**Skills for CommaQA-E task**

Skill <decompose_qa>: Decompose a complex question into a set of sub-questions.
For example: Decompose the question "What awards have movies produced by people born in 1910 won?" into the following sub-questions:
Q1: Who were born in the year 1910?
Q2: Which movies did [A1] produce?
Q3: Which awards were given to [A2]?
Decompose the question "What movies have people from the country Stridery acted in?" into
the following sub-questions: Q1: Who is from the country Stridery?
Q2: Which movies did [A1] act in?

Skill <answer_simple_question>: Answer simple questions about the passage.
For example:
[A Passage]

Q: Which awards were given to Zalate?
A: movie: Zalate ; awarded: Hallowcock. ["Hallowcock"]
Q: Which movies were given the Hallowcock award?
A: movie: Zalate ; awarded: Hallowcock. movie: SkirtSiCine ; award: Hallowcock. ["Zalate",
"SkirtSiCine"]
Q: Which movies did Muntaril direct?
A: movie: Premercy ; directed by: Muntaril. ["Premercy]
Q: Which movies did Muntrail produce?
A: Muntaril produced the movie Premercy with others. Muntaril produced the movie SkirtSiCine
with others. ["Premercy", "SkirtSiCine"]
Q: Which movies did Muntrail write?
A: Muntaril was one of the writers for the movie Zalate. Muntaril wrote for the movie Featsaw. ["Zalate", "Featsaw"]
Q: Who are the actors in the movie Premercy?
A: Monsterscar was an actor in the movie Premercy. ["Monsterscar"]
Q: When was the moive Featsaw released?
A: Monsterscar was an actor in the movie Premercy. ["1973"]

Figure 12: The skills in Skills-in-Context prompt for the CommaQA-E task.

---

**An Example of Skill Composition for the CommaQA-E Task**

**Example**: [A Passage]
Q: What movies have people from the country Stridery acted in?

**Answer**:
1. Using Skill <decompose_qa>, decompose the question "What movies have people from the country Stridery acted in?" into the following sub-questions:

    Q1: Who is from the country Stridery?
    Q2: Which movies did [A1] act in?

2. Using Skill <answer_simple_question>, answer Q1: Who is from the country Stridery?

    Gastrat grew up in the nation of Stridery. A1=["Gastrat"]

3. A1=["Gastrat"], answer Q2: Which movies did [A1] act in?

    i. A1[0]="Gastrat", Using Skill <answer_simple_question>, answer Q2.1: Which movies did Gastrat act in?
      A2.1: Gastrat was an actor in the movie Partnershipmaker. Gastrat was an actor in the movie Partnershipmaker. Gastrat acted in the movie Warpstone. A2.1=["Partnershipmaker", "Nilitude", "Warpstone"]

    ii.So A2=[A2.1]=["Partnershipmaker", "Nilitude", "Warpstone"]

4. The answer is ["Partnershipmaker", "Nilitude", "Warpstone"]

---

Figure 13: An examplar of skill composition in Skills-in-Context prompting for the CommaQA-E task.

---

**Skills for Dynamic Programming**

Skill <list_length>: Get the number of elements in a list.
For example, D=[1,2,3], len(D)=3. A=[1,2,3,4,5,6], len(A)=6.

Skill <find_max>: Find the max number in a list.
For example, max(2,0,-1)=2. max(-3,-2,-1)=-1. max(5,-2,0)=5. max(0,-2,4)=4. max(0,0,0)=0. max(-2,-1,0)=0.

Skill <add>: Add two numbers.
For example, 1+2=3. -2+1=-1. 0+(-3)=-3. 1+(-1)=0.

---

Figure 14: The skills in Skills-in-Context prompt for the task of dynamic programming.

An Example of Skill Composition for the Dynamic Programming Task

**Example**: Compose these skills to do the task: Given a sequence of integers, find a subsequence with the highest sum, such that no two numbers in the subsequence are adjacent in the original sequence.
A=[3,2,1,5].

**Answer**:
1. A=[3,2,1,5] (A[0]=3,A[1]=2,A[2]=1,A[3]=5). Using Skill <list_length>, len(A)=4. Construct a list D=[0,0,0,0] with the len(A)=4 zeros.

2. Update the last element in D. A[3]=5. Using Skill <find_max>, max(5,0)=5. D[3]=5 and D=[0,0,0,5].

3. Update the second last element in D. A[3] = 5, A[2] = 1. Using Skill <find_max>, max(5,1,0) = 5. D[2] = 5 and D=[0,0,5,5].

4. Starting from the third last (len(A)-2=2) element in D, update each element in D based on A=[3,2,1,5] (A[0]=3,A[1]=2,A[2]=1,A[3]=5):

    i. D[3]=5, D[2]=5, A[1]=2. Using Skill <add>, A[1]+D[3]=2+5=7. Using Skill <find_max>, max(D[2],A[1]+D[3],0)=max(5,7,0)=7. D[1]=7 and D=[0,7,5,5].
    ii. D[2]=5, D[1]=7, A[0]=3. Using Skill <add>, A[0]+D[2]=3+5=8. Using Skill <find_max>, max(D[1],A[0]+D[2],0)=max(7,8,0)=8. D[0]=8 and D=[8,7,5,5].

5. D=[8,7,5,5]. The highest sum is D[0]=8.

Figure 15: An examplar of skill composition in Skills-in-Context prompting for the dynamic programming task to find the highest sum of the subsequence.

---

**Skills for GSM8K**

Skill <extract_digits>: Extract the digits in a number to a list. For example, extract digits in 123 to D=[1,2,3]. Extract digits in 7654 to D=[7,6,5,4]

Skill <list_length>: Get the number of elements in a list. For example, D=[1,2,3], len(D)=3. A=[1,2,4,5,6], len(A)=5.

Skill <add_two_single_digit_number>: Add two single-digit numbers. For example, 0+0=0 0+1=1 0+2=2 0+3=3 0+4=4 0+5=5 0+6=6 0+7=7 0+8=8 0+9=9

Skill <sub_two_single_digit_number>: Subtract two single-digit numbers. For example, 0-0=0 0-1=-1 0-2=-2 0-3=-3 0-4=-4 0-5=-5 0-6=-6 0-7=-7 0-8=-8 0-9=-9

Skill <sub_10>: Subtract 10 from a given number. 10-10=0 11-10=1 12-10=2 13-10=3 14-10=4 15-10=5 16-10=6 17-10=7 18-10=8 19-10=9

Skill <add_10>: Add 10 to a given number. -10+10=0 -9+10=1 -8+10=2 -7+10=3 -6+10=4 -5+10=5 -4+10=6 -3+10=7 -2+10=8 -1+10=9

Skill <compare_0>: Compare a number with 0. 10>0 9>0 8>0 7>0 6>0 5>0 4>0 3>0 2>0 1>0 0=0 -1>0 -2>0 -3>0 -4>0 -5>0 -6>0 -7>0 -8>0 -9>0

Skill <compare_10>: Compare a number with 10. 0<10 1<10 2<10 3<10 4<10 5<10 6<10 7<10 8<10 9<10 10=10 11>10 12>10 13>10 14>10 15>10 16>10 17>10 18>10 19>10

Skill <mul_two_single_digit_number>: Multiply two single-digit numbers. For example, 4*1=4 4*2=8 4*3=12 4*4=16 4*5=20 4*6=24 4*7=28 4*8=32 4*9=36

Skill <add_multiple_numbers>: Add multiple numbers such as m+n+p:
1. Add the first two numbers m+n and get the result r1=m+n.
2. Add the third number p to r1 and get the result r2=r1+p.

Skill <add>: Use the skills to add two numbers. For example, calculate 86+964 [The steps to perform the add]

Skill <mul>: Use the skills to multiply two numbers. For example, calculate 86*964 [The steps to perform the multiplication]

Skill : Use the skills to subtract a number from another number. For example, calculate 964-86 [The steps to perform the subtraction]

Skill <age>: Describe the age of a person. If a person is P years old, Q years ago, the person was P-Q years old. If a person is P years old, in Q years, the person will be P+Q years old. If person A is P years old, person B is Q years old, and person A is R years older than person B, then P=Q+R. If person A is P years old, person B is Q years old, and person A is R years younger than person B, then P=Q-R.

Skill <solve_equation>: Solve an equation. When subtracting or adding the same number from both sides of the equation, the equation is still true. When moving a number from one side of the equation to the other side, the sign of the number changes. When moving a multiplication from one side of the equation to the other side, the sign of the multiplication changes. When moving a division from one side of the equation to the other side, the sign of the division changes.
For example, if the equation is 3x+5=7, move +5 to the other side, then 3x=7-5. Using the Skill , 3x=7-5=2. Move 3 to the other side, then x=2/3.

Figure 16: The skills in Skills-in-Context prompt for GSM8K.

> **An Example of Skill Composition for GSM8K**
>
> **Example**: Four years ago, Kody was only half as old as Mohamed. If Mohamed is currently twice 30 years old, how old is Kody?
>
> **Answer**:
> 1. Mohamed is currently twice 30 years old. Using the Skill <mul>, Mohamed is currently 30*2 = 60 years old.
>
> 2. Using Skill <age>, four years ago, Mohamed was 4 years younger than now. Using the Skill , Mohamed was 60-4 = 56 years old.
>
> 3. Four years ago, Kody was only half as old as Mohamed. Using the Skill <div>, Kody was 56/2 = 28 years old.
>
> 4. Using Skill <age>, currently, Kody is 4 years older than four years ago. Using the Skill <add>, Kody is currently 28+4 = 32 years old.
>
> 5. The answer is 32.

Figure 17: An examplar of skill composition in Skills-in-Context prompting for GSM8K math problems.

---

## Skills for MATH

You have the knowledge of many skills, the following are some examples:
Skill <Add>: Add two numbers. For example, 128+987=1115.

Skill : Subtract a number from another number. For example, 128-67=61.

Skill <Mul>: Multiply two numbers. For example, 128*76=9728.

Skill <Div>: Divide a number from another number. For example 12/3=4.

Skill <Mod>: Modulus or modulo, it finds the remainder of a division operation. For example, 10 mod 3 = 1, because 10 divided by 3 leaves a remainder of 1.

Skill <Exp>: An exponent refers to the number of times a number is multiplied by itself. [More Details]

Skill <Base Conversion>: Base conversion is a way to change numbers from one base to another. [More Details]

Skill <Radicals>: A radical represents the root of a number. The square root (represented by sqrt) is the most common radical. [More Details]

Skill <Factoring>: In the context of integers, factorization involves expressing a number as the product of prime numbers. [More Details]

Skill <Solve Equation>: Solve an equation. [More Details]

Skill <Quadratic Formula>: The quadratic formula is used to solve quadratic equations. [More Details]

Skill <Complex Number>: The quadratic formula is used to solve quadratic equations. [More Details]

Skill <Piecewise Function: Continuous>: A piecewise function is continuous if it is continuous at every point in its domain. [More Details]

Skill <Factorial>: Factorial is a function that multiplies a given number by every number below it until 1. [More Details]

Skill <Probability>: Probability is the measure of the likelihood that an event will occur. [More Details]

Skill <Conditional Probability>: The probability of an event occurring given that another event has already occurred. [More Details]

Skill <Probability Addition Rule>: The Addition Rule in probability is used to calculate the probability of either of two events happening. [More Details]

Skill <Probability Multiplication Rule>: A way to determine the probability of two events occurring at the same time (conjointly). [More Details]

Skill <Counting Principle>: If there are m ways to do one thing, and n ways to do another, then there are m*n ways of doing both. [More Details]

Skill <Permutations>: Permutations refer to the arrangement of items in a specific order. [More Details]

Skill <Combination>: Combinations refer to the selection of items without regard to the order. [More Details]

Skill <Perimeter>: The perimeter of a shape is the distance around its boundary. [More Details]

Skill <Area>: The area of a shape is the amount of space that it covers. [More Details]

Skill <Volume>: Volume is the measure of the amount of space that a three-dimensional object occupies. [More Details]

Skill <Prime Numbers>: A prime number is a natural number greater than 1 that has no positive divisors other than 1 and itself. [More Details]

Skill <Composite Numbers>: Composite numbers are positive integers that have at least one positive divisor other than one or the number itself. [More Details]

Skill <GCD>:The Greatest Common Divisor (GCD), also known as the Greatest Common Factor (GCF), [More Details]

Skill <LCM>:The Least Common Multiple (LCM) of two integers is the smallest positive integer that is divisible by both numbers without leaving a remainder. [More Details]

Skill <Congruences>: Two integers a and b are said to be congruent modulo n if they have the same remainder when divided by n. [More Details]

Figure 18: The skills in Skills-in-Context prompt for MATH.

**An Example of Skill Composition for MATH**

**Example**: Shown below are rows 1, 2, and 3 of Pascal's triangle.[Pascal's triangle]. Let $(a_i)$, $(b_i)$, $(c_i)$ be the sequence, from left to right, of elements in the 2005th, 2006th, and 2007th rows, respectively, with the leftmost element occurring at $i = 0$. Compute $[\sum_{i=0}^{2006} \frac{b_i}{c_i} - \sum_{i=0}^{2005} \frac{a_i}{b_i}.]$

**Answer**:
1. Using the Skill <Pascal's Triangle>, the number in the n-th row and k-th column of the Pascal's triangle is C(n,k).

2. Suppose $(a_i), (b_i), (c_i)$ represent the entries in rows n - 1, n, n + 1 of Pascal's triangle. $a_i = C(n-1, i), b_i = C(n, i), c_i = C(n+1, i)$

3. Using the Skill <Combination>, $\sum_{i=0}^{2006} \frac{b_i}{c_i} = \sum_{i=0}^{2006} \frac{\binom{2006}{i}}{\binom{2007}{i}} = \sum_{i=0}^{2006} \frac{2007-i}{2007} = 2007 - \frac{2006 \cdot 2007}{2 \cdot 2007} = 2007 - (2006 * 2007)/(2 * 2007) = 2007 - 1003 = 1004$

4. Using the Skill <Combination>, $\sum_{i=0}^{2005} \frac{a_i}{b_i} = \sum_{i=0}^{2005} \frac{\binom{2005}{i}}{\binom{2006}{i}} = \sum_{i=0}^{2005} \frac{2006-i}{2006} = 2006 - \frac{2005 \cdot 2006}{2 \cdot 2006} = 2006 - (2005 * 2006)/(2 * 2006) = 2006 - 1002.5 = 1003.5$

5. Using the Skill , $[\sum_{i=0}^{2006} \frac{b_i}{c_i} - \sum_{i=0}^{2005} \frac{a_i}{b_i}] = 1004 - 1003.5 = 0.5$

6. The answer is 0.5.

Figure 19: An examplar of skill composition in Skills-in-Context prompting for MATH problems.

# E APPENDIX: GENERATION EXAMPLES

We further share some example generation from ChatGPT with our Skills-in-Context prompts on all the tasks in Figure 20,21, 22, 23, 24, 25, 26, 27, 28, 29, 30, 31, 32, 33.

---

**Last Letter Concatenation**

**Question**: Take the last letters of the words in 'than, appropriate, sees, daisy, disturbed, espn, stable, double, luther, shut' and concatenate them.

- - - - - - - - - - - - - - - - - - - - - - - - - - - - - - - - - - - - - - - - - - - - -

**Model Output**:

1. Using the Skill <words_to_list>, put the asked words, 'than, appropriate, sees, daisy, disturbed, espn, stable, double, luther, shut', to a list. D=['than', 'appropriate', 'sees', 'daisy', 'disturbed', 'espn', 'stable', 'double', 'luther', 'shut']

2. Get the last letter of each word in the list D=['than', 'appropriate', 'sees', 'daisy', 'disturbed', 'espn', 'stable', 'double', 'luther', 'shut'] to a new list R=[]:

    i. Using the Skill <last_letter>, the last letter of D[0]='than' is 'n'. R=[n]
    ii. Using the Skill <last_letter>, the last letter of D[1]='appropriate' is 'e'. R=[n, e]
iii. Using the Skill <last_letter>, the last letter of D[2]='sees' is 's'. R=[n, e, s]
    iv. Using the Skill <last_letter>, the last letter of D[3]='daisy' is 'y'. R=[n, e, s, y]
    v. Using the Skill <last_letter>, the last letter of D[4]='disturbed' is 'd'. R=[n, e, s, y, d]
    vi. Using the Skill <last_letter>, the last letter of D[5]='espn' is 'n'. R=[n, e, s, y, d, n]
    vii. Using the Skill <last_letter>, the last letter of D[6]='stable' is 'e'. R=[n, e, s, y, d, n, e]
    viii. Using the Skill <last_letter>, the last letter of D[7]='double' is 'e'. R=[n, e, s, y, d, n, e, e]
    ix. Using the Skill <last_letter>, the last letter of D[8]='luther' is 'r'. R=[n, e, s, y, d, n, e, e, r]
    x. Using the Skill <last_letter>, the last letter of D[9]='shut' is 't'. R=[n, e, s, y, d, n, e, e, r, t]

3. R=[n, e, s, y, d, n, e, e, r, t]. The answer is 'n e s y d n e e r t'.

---

Figure 20: An example of the generated answer on last-letter-concatenation task using ChatGPT with our Skills-in-Context prompting.

---

**Addition**

**Question**: Calculate 165789+876534

- - - - - - - - - - - - - - - - - - - - - - - - - - - - - - - - - - - - - - - - - - - - - - - - -

**Model Output**:

1. Using Skill <extract_digits> and Skill <list_length>, extract the digits in 165789 to DM=[1,6,5,7,8,9]. len(DM)=6. Extract the digits in 876534 to DN=[8,7,6,5,3,4]. len(DN)=6.

2. len(DM)=6=len(DN)=6. Align the two lists: DM=[1,6,5,7,8,9], DN=[8,7,6,5,3,4].

3. Using Skill <add_two_single_digit_number>, add every number in DM=[1,6,5,7,8,9] to the corresponding number in DN =[8,7,6,5,3,4] to form a list, DS=[1+8,6+7,5+6,7+5,8+3,9+4]=[9,13,11,12,11,13].

4. Append a 0 to the beginning of DS: DS=[0,9,13,11,12,11,13]. Construct a list R=[0,0,0,0,0,0,0] with len(DS)=7 zeros.

5. DS=[0,9,13,11,12,11,13] (DS[0]=0, DS[1]=9, DS[2]=13, DS[3]=11, DS[4]=12, DS[5]=11, DS[6]=13), starting from DS[6] to DS[1] (DS[6]=13, DS[5]=11, DS[4]=12, DS[3]=11, DS[2]=13, DS[1]=9):

    i. DS[6]=13, R[6]=0, R[6]=R[6]+DS[6]=0+13=13. Based on Skill <compare_10>, 13>10, R[6]=13>10, so R[5]=1 and R[6]=13-10=3 by Skill <sub_10>. R=[R[0],R[1],R[2],R[3],R[4],R[5],R[6]]=[0,0,0,0,0,1,3].
    ii. DS[5]=11, R[5]=1, R[5]=R[5]+DS[5]=1+11=12. Based on Skill <compare_10>, 12>10, R[5]=12>10, so R[4]=1 and R[5]=12-10=2 by Skill <sub_10>. R=[R[0],R[1],R[2],R[3],R[4],R[5],R[6]]=[0,0,0,0,1,2,3].
    iii. DS[4]=12, R[4]=1, R[4]=R[4]+DS[4]=1+12=13. Based on Skill <compare_10>, 13>10, R[4]=13>10, so R[3]=1 and R[4]=13-10=3 by Skill <sub_10>. R=[R[0],R[1],R[2],R[3],R[4],R[5],R[6]]=[0,0,0,1,3,2,3].
    iv. DS[3]=11, R[3]=1, R[3]=R[3]+DS[3]=1+11=12. Based on Skill <compare_10>, 12>10, R[3]=12>10, so R[2]=1 and R[3]=12-10=2 by Skill <sub_10>. R=[R[0],R[1],R[2],R[3],R[4],R[5],R[6]]=[0,0,1,2,3,2,3].
    v. DS[2]=13, R[2]=1, R[2]=R[2]+DS[2]=1+13=14. Based on Skill <compare_10>, 14>10, R[2]=14>10, so R[1]=1 and R[2]=14-10=4 by Skill <sub_10>. R=[R[0],R[1],R[2],R[3],R[4],R[5],R[6]]=[0,1,4,2,3,2,3].
    vi. DS[1]=9, R[1]=1, R[1]=R[1]+DS[1]=1+9=10. Based on Skill <compare_10>, 10=10, R[1]=10=10, so R[0]=1 and R[1]=10-10=0 by Skill <sub_10>. R=[R[0],R[1],R[2],R[3],R[4],R[5],R[6]]=[1,0,4,2,3,2,3].
6. R=[1,0,4,2,3,2,3]. The answer is 1042323.

Figure 21: An example of the generated answer on the addition task using ChatGPT with Skills-in-Context prompting.

---

**Multiplication**

**Question**: Calculate 1467*7651

- - - - - - - - - - - - - - - - - - - - - - - - - - - - - - - - - - - - - - - - - - -

**Model Output**:

1. Using Skill <extract_digits> and Skill <list_length>, extract the digits in 1467 to DM=[1,4,6,7]. len(DM)=4. Extract the digits in 7651 to DN=[7,6,5,1]. len(DN)=4.

2. Add 0,1,len(DM)-1=3 zeros to the end of every number in DM=[1,4,6,7] according to the position of the number in DM: DMO=[1*1000,4*100,6*10,7*1]=[1000,400,60,7].

3. Add 0,len(DN)-1=3 zeros to the end of every number in DN=[7,6,5,1] according to the position of the number in DN: DNO=[7*1000,6*100,5*10,1*1]=[7000,600,50,1].

4. Using Skill <mul_two_single_digit_number>, multiple every number in DMO=[1000,400,60,7] with every number in DNO=[7000,600,50,1] and get R=[1000*7000,1000*600,1000*50,1000*1, 400*7000,400*600,400*50,400*1,60*7000,60*600,60*50,60*1,7*7000,7*600,7*50,7*1]= [7000000,600000,50000,1000,2800000,240000,20000,400,420000,36000,3000,60, 49000,4200,350,7].

5. Using Skill <add_multiple_numbers>, add all the numbers in R=[7000000,600000,50000,1000, 2800000,240000,20000,400,420000,36000,3000,60, 49000,4200,350,7], 7000000+600000+50000+1000+2800000+240000+20000+400+420000 +36000+3000+60+49000+4200+350+7:

    i. Add the first two numbers: r1=7000000+600000=7600000.
    ii. Add the third number 50000 to r1=7600000: r2=7600000+50000=7650000.
    iii. Add the fourth number 1000 to r2=7650000: r3=7650000+1000=7651000.
    iv. Add the fifth number 2800000 to r3=7651000: r4=7651000+2800000=10451000.
    v. Add the sixth number 240000 to r4=10451000: r5=10451000+240000=10691000.
    vi. Add the seventh number 20000 to r5=10691000: r6=10691000+20000=10711000.
    vii. Add the eighth number 400 to r6=10711000: r7=10711000+400=10711400.
    viii. Add the ninth number 420000 to r7=10711400: r8=10711400+420000=11131400.
    ix. Add the tenth number 36000 to r8=11131400: r9=11131400+36000=11167400.
    x. Add the eleventh number 3000 to r9=11167400: r10=11167400+3000=11170400.
    xi. Add the twelfth number 60 to r10=11170400: r11=11170400+60=11170460.
    xii. Add the thirteenth number 49000 to r11=11170460: r12=11170460+49000=11219460.
    xiii. Add the fourteenth number 4200 to r12=11219460: r13=11219460+4200=11223660.
    xiv. Add the fifteenth number 350 to r13=11223660: r14=11223660+350=11224010.
    xv. Add the sixteenth number 7 to r14=11224010: r15=11224010+7=11224017.

6. So the answer is 11224017

Figure 22: An example of the generated answer on the multiplication task using ChatGPT with Skills-in-Context prompting.

---

**CommaQA-E**

**Question**:

movie: Skob ; directed by: Haldron. movie: Riften ; directed by: Conanopeia. movie: Tetroxidine ; directed by: Conanopeia. movie: Sheepcrest ; directed by: Flumph. movie: Sheepcrest ; award: Mariskenna. movie: Skob ; award: Jubeus. movie: Tetroxidine ; award: Handt. movie: Riften ; awarded: Handt. movie: Tetroxidine ; written by: Conanopeia. movie: Riften ; written by: Conanopeia. movie: Riften ; writer: Flumph. movie: Tetroxidine ; written by: Haldron. movie: Skob ; written by: Haldron. movie: Sheepcrest ; writer: Flumph. movie: Skob ; year: 1962. movie: Tetroxidine ; release year: 1997. movie: Riften ; release year: 1972. movie: Sheepcrest ; release year: 1997. Flumph was an actor in the movie Riften. Flumph acted in the movie Sheepcrest. Conanopeia was an actor in the movie Riften. Conanopeia acted in the movie Tetroxidine. Conanopeia acted in the movie Sheepcrest. Haldron acted in the movie Tetroxidine. Haldron acted in the movie Skob. Haldron was born in the year 1939. Flumph was born in the year 1938. Conanopeia was born in 1938. Haldron is from the country of Poquet. Conanopeia grew up in the nation of Poquet. Flumph grew up in the nation of Stridery. Diaqum was awarded to Flumph. Lidus was awarded to Conanopeia. Haldron won the Lidus award. Conanopeia was one of the producers of the movie Tetroxidine. Flumph was one of the producers of the movie Sheepcrest. Flumph produced the movie Skob with others. Haldron was one of the producers of the movie Sheepcrest. Haldron was one of the producers of the movie Riften. Haldron produced the movie Skob with others. Conanopeia was one of the producers of the movie Riften.

Q: What movies have the people from Poquet written?

- - - - - - - - - - - - - - - - - - - - - - - - - - - - - - - - - - - - - - - - -

**Model Output**:

1. Using Skill <decompose_qa>, decompose the question "What movies have the people from Poquet written?" into the following sub-questions:

    Q1: Who is from the country Poquet?
    Q2: Which movies did [A1] write?

2. Using Skill <answer_simple_question>, answer Q1: Who is from the country Poquet? Conanopeia grew up in the nation of Poquet. Haldron is from the country of Poquet. A1=["Conanopeia", "Haldron"]
3. A1=["Conanopeia", "Haldron"]. Using Skill <answer_simple_question>, answer Q2: Which movies did [A1] write?

    i. A1[0]="Conanopeia", answer Q2.1: Which movies did Conanopeia write?
        A2.1: movie: Tetroxidine ; written by: Conanopeia. movie: Riften ; written by: Conanopeia. A2.1=["Tetroxidine", "Riften"]
    ii. A1[1]="Haldron", answer Q2.2: Which movies did Haldron write?
        A2.2: movie: Skob ; written by: Haldron. A2.2=["Skob"]
    iii. So A2=A2.1+A2.2=["Tetroxidine", "Riften", "Skob"]

4. The answer is ["Tetroxidine", "Riften", "Skob"]

---

Figure 23: An example of the generated answer on the CommaQA-E task using ChatGPT with our Skills-in-Context prompting.

**DP**

**Question**: Given a sequence of integers, find a subsequence with the highest sum, such that no two numbers in the subsequence are adjacent in the original sequence.
A=[1, -5, 3, -4, -1, -1, 2, 2].

**Model Output**:

1. A=[1, -5, 3, -4, -1, -1, 2, 2] (A[0]=1,A[1]=-5,A[2]=3,A[3]=-4,A[4]=-1,A[5]=-1,A[6]=2,A[7]=2). Using Skill <list_length>, len(A)=8. Construct a list D=[0,0,0,0,0,0,0,0] with the len(A)=8 zeros.

2. Update the last element in D. A[7]=2. Using Skill <find_max>, max(2,0)=2. D[7]=2 and D=[0,0,0,0,0,0,0,2].

3. Update the second last element in D. A[7]=2, A[6]=2. Using Skill <find_max>, max(2,2,0)=2. D[6]=2 and D=[0,0,0,0,0,0,2,2].

4. Starting from the third last (len(A)-2=6) element in D, update each element in D based on A=[1, -5, 3, -4, -1, -1, 2, 2] (A[0]=1,A[1]=-5,A[2]=3,A[3]=-4, A[4]=-1,A[5]=-1,A[6]=2,A[7]=2):

    i. D[7]=2, D[6]=2, A[5]=-1. Using Skill <add>, A[5]+D[7]=-1+2=1. Using Skill <find_max>, max(D[6],A[5]+D[7],0)=max(2,1,0)=2. D[5]=2 and D=[0,0,0,0,0,2,2,2].
    ii. D[6]=2, D[5]=2, A[4]=-1. Using Skill <add>, A[4]+D[6]=-1+2=1. Using Skill <find_max>, max(D[5],A[4]+D[6],0)=max(2,1,0)=2. D[4]=2 and D=[0,0,0,0,2,2,2,2].
    iii. D[5]=2, D[4]=2, A[3]=-4. Using Skill <add>, A[3]+D[5]=-4+2=-2. Using Skill <find_max>, max(D[4],A[3]+D[5],0)=max(2,-2,0)=2. D[3]=2 and D=[0,0,0,2,2,2,2,2].
    iv. D[4]=2, D[3]=2, A[2]=3. Using Skill <add>, A[2]+D[4]=3+2=5. Using Skill <find_max>, max(D[3],A[2]+D[4],0)=max(2,5,0)=5. D[2]=5 and D=[0,0,5,2,2,2,2,2].
    v. D[3]=2, D[2]=5, A[1]=-5. Using Skill <add>, A[1]+D[3]=-5+2=-3. Using Skill <find_max>, max(D[2],A[1]+D[3],0)=max(5,-3,0)=5. D[1]=5 and D=[0,5,5,2,2,2,2,2].
    vi. D[2]=5, D[1]=5, A[0]=1. Using Skill <add>, A[0]+D[2]=1+5=6. Using Skill <find_max>, max(D[1],A[0]+D[2],0)=max(5,6,0)=6. D[0]=6 and D=[6,5,5,2,2,2,2,2].

5. D=[6,5,5,2,2,2,2,2]. The highest sum is D[0]=6.

Figure 24: An example of the generated answer on the dynamic programming task using ChatGPT with our Skills-in-Context prompting.

---

**GSM8K**

**Question**: The girls are trying to raise money for a carnival. Kim raises $320 more than Alexandra, who raises $430, and Maryam raises $400 more than Sarah, who raises $300. How much money, in dollars, did they all raise in total?

- - - - - - - - - - - - - - - - - - - - - - - - - - - - - - - - - - - - - - - - - - - - - - - - - - -

**Model Output**:

1. Alexandra raises $430.

2. Kim raises $320 more than Alexandra. Using the Skill <add>, Kim raises $430+$320=$750.

3. Sarah raises $300.

4. Maryam raises $400 more than Sarah. Using the Skill <add>, Maryam raises $300+$400=$700.

5. Using the Skill <add>, they all raise $430+$750+$300+$700=$2180 in total.

6. The answer is 2180.

---

Figure 25: An example of the generated answer on the GSM8K task using ChatGPT with Skills-in-Context prompting.

---

**GSM8K**

**Question**: It's strawberry-picking time on Grandma Concetta's farm. Tony can pick 6 quarts of strawberries per hour, while Bobby picks one less quart of strawberries per hour than Tony. Kathy can pick twice as many strawberries per hour as Bobby, and Ricky picks two fewer quarts of strawberries per hour than does Kathy. In total, how many quarts of strawberries can Tony, Bobby, Ricky, and Kathy pick per hour on Grandma Concetta's farm?

- - - - - - - - - - - - - - - - - - - - - - - - - - - - - - - - - - - - - - - - - - - - - - - - - -

**Model Output**:
1. Tony can pick 6 quarts of strawberries per hour.

2. Bobby picks one less quart of strawberries per hour than Tony. Using the Skill , Bobby picks 6-1=5 quarts of strawberries per hour.

3. Kathy can pick twice as many strawberries per hour as Bobby. Using the Skill <mul>, Kathy picks 5*2=10 quarts of strawberries per hour.

4. Ricky picks two fewer quarts of strawberries per hour than does Kathy. Using the Skill , Ricky picks 10-2=8 quarts of strawberries per hour.

5. In total, Tony, Bobby, Ricky, and Kathy can pick 6+5+10+8 quarts of strawberries per hour. Using the Skill <add_multiple_numbers>:

    i. Add the first two numbers using Skill <add>: r1=6+5=11.
    ii. Add the third number 10 to r1=11 using Skill <add>: r2=11+10=21.
    iii. Add the fourth number 8 to r2=21 using Skill <add>: r3=21+8=29.

6. So the answer is 29.

---

Figure 26: An example of the generated answer on the GSM8K task with our Skills-in-Context prompting, where <add_multiple_numbers> is included as a basic skill in the SKiC prompting context (see Table 16) but is not demonstrated in any given examplar to show how to use it. LLMs automatically figure out how to use such skills in an innovative composition to solve an unseen complex problem.

---

**GSM8K**

**Question**: A merchant wants to make a choice of purchase between 2 purchase plans: jewelry worth $5,000 or electronic gadgets worth $8,000. His financial advisor speculates that the jewelry market will go up 2.5% while the electronic gadgets market will rise 1.2% within the same month. If the merchant is looking to maximize profit at the end of this month by making a choice, how much profit would this be?

- - - - - - - - - - - - - - - - - - - - - - - - - - - - - - - - - - - - - - - - - - - - - - - -

**Model Output**:

1. If the merchant buys jewelry worth $5,000 and the jewelry market goes up 2.5%, using the Skill <mul>, the value of the jewelry will increase by $5,000*2.5%= $125. Using Skill <add>, the value of the jewelry will be $5,000+$125=$5125.

2. If the merchant buys electronic gadgets worth $8,000 and the electronic gadgets market goes up 1.2%, using the Skill <mul>, the value of the electronic gadgets will increase by $8,000*1.2%= $96. Using Skill <add>, the value of the electronic gadgets will be $8,000+$96=$8096.

3. The merchant wants to maximize profit. Using the Skill , the profit from buying jewelry will be $5125-$5000=$125. The profit from buying electronic gadgets will be $8096-$8000=$96.

4. Using the Skill <compare>, $125>$96, so the merchant should buy jewelry to maximize profit. The profit will be $125.

5. The answer is 125.

---

Figure 27: An example of the generated answer on the GSM8K task with our Skills-in-Context prompting, where the skill <compare> are neither included in the SKiC prompting context (see Table 16) nor used in any given examplars. LLMs utilize the skills pre-existing in their pre-trained knowledge to solve the problem.

---

**GSM8K**

**Question**: Mr. Jackson's fourth-grade class has 27 students. He wants to give each student 2 glue sticks. The glue sticks come in packs of 8. How many packs will Mr. Jackson need to buy so every student can have 2 glue sticks, assuming he can only buy whole packs and he expects to have some extra glue sticks left over?

- - - - - - - - - - - - - - - - - - - - - - - - - - - - - - - - - - - - - - -

**Model Output**:

1. Mr. Jackson's fourth-grade class has 27 students and he wants to give each student 2 glue sticks. Using the Skill <mul>, Mr. Jackson needs 27*2=54 glue sticks.

2. The glue sticks come in packs of 8. Using the Skill <div>, Mr. Jackson needs 54/8=6.75 packs of glue sticks.

3. Mr. Jackson can only buy whole packs. Using the Skill <round>, Mr. Jackson needs to buy 7 packs of glue sticks.

4. The answer is 7.

---

Figure 28: An example of the generated answer on the GSM8K task with our Skills-in-Context prompting, where the skill <round> are neither included in the original SKiC prompting context (see Table 16) nor used in any given examplars. LLMs utilize the skills pre-existing in their pre-trained knowledge to solve the problem.

---

**MATH**

**Question**: What is the smallest possible average of four distinct positive even integers?

- - - - - - - - - - - - - - - - - - - - - - - - - - - - - - - - - - - - - - - - - - - - - - -

**Model Output**:

1. The smallest possible average of four distinct positive even integers is achieved when the four integers are the smallest possible.

2. The smallest four distinct positive even integers are 2, 4, 6, and 8.

3. Using the Skill $<$Average$>$, the average of these four numbers is (2+4+6+8)/4=20/4=5.

4. The answer is 5.

---

Figure 29: An example of the generated answer on the MATH task with our Skills-in-Context prompting, where the skill $<$Average$>$ are neither included in the original SKiC prompting context (see Table 18) nor used in any given examplars. LLMs(GPT4) utilize the skills pre-existing in their pre-trained knowledge to solve the problem.

---

**MATH**

**Question**: How many numbers are in the list $2008, 2003, 1998, \ldots, 8, 3$?

- - - - - - - - - - - - - - - - - - - - - - - - - - - - - - - - - - - - - - - - - - - - - -

**Model Output**:

1. The list starts at 2008 and ends at 3, and the difference between consecutive terms is 5.

2. Using the Skill $\boxed{\text{<Arithmetic Sequence>}}$, the n-th term of an arithmetic sequence can be found using the formula $a_n = a_1 + (n-1)d$, where $a_n$ is the nth term, $a_1$ is the first term, $n$ is the term number, and $d$ is the common difference.

3. Substitute $a_n$ with 3, $a_1$ with 2008, and d with $-5$ into the formula, $3 = 2008 + (n-1)(-5)$.

4. Using the Skill <Solve Equation>, move 2008 to the other side, $(n-1)(-5) = 3-2008$. Using the Skill , $(n-1)(-5) = -2005$.

5. Using the Skill <Solve Equation>, move $*(-5)$ to the other side, $n-1 = -2005/-5$. Using the Skill <Div>, $n-1 = 401$.

6. Using the Skill <Solve Equation>, move $-1$ to the other side, $n = 401 + 1$. Using the Skill <Add>, $n = 402$.

7. The answer is 402.

Figure 30: An example of the generated answer on the MATH task with our Skills-in-Context prompting, where the skill <Arithmetic Sequence> are neither included in the original SKiC prompting context (see Table 18) nor used in any given examplars. LLMs(GPT4) utilize the skills pre-existing in their pre-trained knowledge to solve the problem.

---

**MATH**

**Question**: Suppose $P$ is the point $(5, 3)$ and $Q$ is the point $(-3, 6)$. What is the midpoint of $\overline{PQ}$?

- - - - - - - - - - - - - - - - - - - - - - - - - - - - - - - - - - - - - - - - - - - - - - - - -

**Model Output**:

1. Using the Skill $\boxed{\text{<Midpoint Formula>}}$, the midpoint of two points $(x_1, y_1)$ and $(x_2, y_2)$ is given by $\left(\frac{x_1+x_2}{2}, \frac{y_1+y_2}{2}\right)$.

2. For points $P(5, 3)$ and $Q(-3, 6)$, the midpoint is $\left(\frac{5+(-3)}{2}, \frac{3+6}{2}\right)$.

3. Using the Skill <Add> and <Div>, the midpoint is $\left(\frac{2}{2}, \frac{9}{2}\right) = (1, 4.5)$.

4. The answer is $(1, 4.5)$.

Figure 31: An example of the generated answer on the MATH task with our Skills-in-Context prompting, where the skill <Midpoint Formula> are neither included in the original SKiC prompting context (see Table 18) nor used in any given examplars. LLMs(GPT4) utilize the skills pre-existing in their pre-trained knowledge to solve the problem.

---

**MATH**

**Question**: If **a** and **b** are vectors such that $\|\mathbf{a}\| = 2$, $\|\mathbf{b}\| = 7$, and $\mathbf{a} \times \mathbf{b} = \begin{pmatrix} 3 \\ 2 \\ 6 \end{pmatrix}$, then find the smallest possible angle between **a** and **b**.

- - - - - - - - - - - - - - - - - - - - - - - - - - - - - - - - -

**Model Output**:

1. Using the Skill <Cross Product>, the magnitude of the cross product of two vectors is equal to the product of the magnitudes of the vectors and the sine of the angle between them. So, we have $\|\mathbf{a} \times \mathbf{b}\| = \|\mathbf{a}\|\|\mathbf{b}\| \sin\theta$, where $\theta$ is the angle between **a** and **b**.

2. Using the Skill <Vector Magnitude>, the magnitude of the cross product $\|\mathbf{a} \times \mathbf{b}\|$ is $\sqrt{3^2 + 2^2 + 6^2} = \sqrt{49} = 7$.

3. Substituting the given values, we get $7 = 2 * 7 * \sin\theta$. Using the Skill <Div> and <Mul>, we get $\sin\theta = 7/(2*7) = 1/2$.

4. Using the Skill <Inverse Trigonometric Functions>, the angle whose sine is 1/2 is 30 degrees or 150 degrees. But we want the smallest possible angle, so $\theta = 30$ degrees.

5. The answer is 30.

Figure 32: An example of the generated answer on the MATH task with our Skills-in-Context prompting, where the skill <Cross Product>, <Vector Magnitude>,<Inverse Trigonometric Functions> are neither included in the original SKiC prompting context (see Table 18) nor used in any given examplars. LLMs(GPT4) utilize the skills pre-existing in their pre-trained knowledge to solve the problem.

---

**MATH**

**Question**: Let $\triangle ABC$ be a triangle such that $AB = 13$, $BC = 14$, and $CA = 15$. Meanwhile, $D$ is a point on $BC$ such that $AD$ bisects $\angle A$. Find the area of $\triangle ADC$.

- - - - - - - - - - - - - - - - - - - - - - - - - - - - - - - - - - - - - - - - -

**Model Output**:

1. Using the Skill <Angle Bisector Theorem>, the ratio of the lengths of the two segments created by the angle bisector is equal to the ratio of the lengths of the two opposite sides. So, BD/DC = AB/AC = 13/15.

2. Using the Skill <Solve Equation>, BD = 14*13/(13+15) = 6.5, DC = 14*15/(13+15) = 7.5.

3. Using the Skill <Heron's Formula>, the area of triangle ABC = $\sqrt{s(s-AB)(s-BC)(s-AC)}$, where $s = (AB + BC + AC)/2$ is the semi-perimeter of the triangle. So, $s = (13 + 14 + 15)/2 = 21$, and the area of triangle ABC = $\sqrt{21(21-13)(21-14)(21-15)} = 84$ square units.

4. Using the Skill <Area>, the area of triangle ADC = (Area of triangle ABC) * (DC/BC) = 84 * (7.5/14) = 45 square units.

5. The answer is 45.

Figure 33: An example of the generated solution on the MATH task using our SKiC prompting. The skills <Angle Bisector Theorem> and <Heron's Formula> are neither provided in the SKiC prompting context (see Figure 18) nor used in any given examplars. LLMs harness the internal skills in their pre-trained knowledge to solve the problem.

---

movie: Geissant ; directed by: Carpoon. movie: Myristorrhoid ; directed by: Carpoon. movie: Riften ; directed by: Muntaril. movie: Chimpwurst ; director: Midcareer. movie: Chimpwurst ; awarded: Neuropsychotaxis. movie: Riften ; awarded: Neuropsychotaxis. movie: Geissant ; awarded: Triptychology. movie: Myristorrhoid ; award: Monoxandrite. award: Lameze ; winner: Carpoon. award: Lameze ; winner: Midcareer. award: Modiparity ; winner: Muntaril. movie: Myristorrhoid ; release year: 1974. movie: Chimpwurst ; year: 1962. movie: Geissant ; year: 2019. movie: Riften ; year: 1974. Midcareer acted in the movie Myristorrhoid. Carpoon was an actor in the movie Chimpwurst. Muntaril acted in the movie Geissant. Carpoon was an actor in the movie Chimpwurst. Muntaril acted in the movie Riften. Midcareer was born in 1943. Carpoon was born in 1921. Muntaril was born in 1921. Midcareer is from the country of Whime. Muntaril grew up in the nation of Whime. Carpoon grew up in the nation of Poquet. Midcareer was one of the producers of the movie Riften. Carpoon was one of the producers of the movie Geissant. Carpoon produced the movie Riften with others. Muntaril produced the movie Chimpwurst with others. Carpoon was one of the producers of the movie Myristorrhoid. Midcareer was one of the producers of the movie Myristorrhoid. Muntaril produced the movie Geissant with others. Muntaril wrote for the movie Geissant. Carpoon was one of the writers for the movie Chimpwurst. Midcareer was one of the writers for the movie Myristorrhoid. Muntaril wrote for the movie Riften. Midcareer was one of the writers for the movie Geissant.

*Q: What movies have people from the country Whime acted in?*

| Decomp | SKiC |
|---|---|
| A: Q: (select) [qa] Who is from the country Whime?
  Q: Who is from the country Whime?
  A: ['Midcareer']
Q: (select)
O: select(['Midcareer']) ==> ['Midcareer']
A: ['Midcareer']
A: ['Midcareer']
Q: (project_values_flat_unique) [qa] Which movies has #1 been an actor in?
  Q: Which movies has Midcareer been an actor in?
  A: ['Myristorrhoid', 'Chimpwurst']
Q: (project_values_flat_unique) ['Midcareer']
O: project_values_flat_unique([['Midcareer', ['Myristorrhoid', 'Chimpwurst']]]) ==> ['Myristorrhoid', 'Chimpwurst']
  A: ['Myristorrhoid', 'Chimpwurst']
A: ['Myristorrhoid', 'Chimpwurst']
Q: [EOQ] | 1. Using the Skill <decompose_qa>, decompose the question 'What movies have people from the country Whime acted in?' into the following sub-questions:
  Q1: Who is from the country Whime?
  Q2: Which movies did [A1] act in?
2. Using the Skill <answer_simple_question>, answer Q1: Who is from the country Whime?
  Midcareer is from the country of Whime. Muntaril grew up in the nation of Whime. A1=['Midcareer', 'Muntaril']
3. A1=['Midcareer', 'Muntaril']. Using the Skill <answer_simple_question>, answer Q2: Which movies did [A1] act in?
  i. A1[0]='Midcareer', answer Q2.1: Which movies did Midcareer act in? A2.1: Midcareer acted in the movie Myristorrhoid. Midcareer acted in the movie Chimpwurst. ['Myristorrhoid', 'Chimpwurst']
  ii. A1[1]='Muntaril', answer Q2.2: Which movies did Muntaril act in? A2.2: Muntaril acted in the movie Geissant. Muntaril acted in the movie Riften. ['Geissant', 'Riften']
  iii. So A2=A2.1+A2.2=['Myristorrhoid', 'Chimpwurst', 'Geissant', 'Riften']
4. The answer is ['Myristorrhoid', 'Chimpwurst', 'Geissant', 'Riften'] |

Figure 34: Examples of the generated answers from Decomp prompting and our SKiC prompting, respectively. The correct answer should be "Myristorrhoid, Chimpwurst, Geissant, Riften". Errors (highlighted in red) in early stages in Decomp propagate to final incorrect answers while our SKiC avoids such errors (highlighted in green).

## F  APPENDIX: THE MOST FREQUENTLY USED SKILLS BY GPT-4 FOR SOLVING MATH BENCHMARK

In Table 12, we report the most frequently used skills by GPT-4 to solve the MATH problems. There are two sources of the skills: (i) the ones provided in the context of SKiC prompts, and (ii) the ones originating from GPT-4's internal knowledge (acquired through pretraining).

Table 12: The most frequently used skills by GPT-4 for solving MATH benchmark with SKiC prompting. The skills can be from the context of the SKiC prompts (denoted as "in-context" in the table) or from the internal knowledge acquired during the pretraining stage (denoted as "internal").

| Category | Source | Top Used Skills |
|---|---|---|
| Pre-Algebra | In-context | Div, Mul, Add, Sub, Solve Equation, Area, Exp, Counting Principle, Radicals, Prime Numbers |
| | Internal | Pythagorean Theorem, Rounding, Divisibility Rules, Percentage, Angles, Simply Fraction, Mean, Ratio, Triangle Angle Sum, Order of Operations |
| Geometry | In-context | Area, Mul, Div, Add, Sub, Solve Equation, Volume, Radicals, Exp, Perimeter |
| | Internal | Pythagorean Theorem, Trigonometry, Triangle, Triangle Inequality, Similar Triangles, Circle, Geometry, Triangle Angle Sum, Angle Bisector Theorem, Trigonometric Ratios |
| Inter-Algebra | In-context | Factoring, Solve Equation, Add, Mul, Sub, Complex Number, Inequality, Quadratic Formula, Div, Exp |
| | Internal | Substitution, Completing the Square, Polynomial, Logarithm, AM-GM Inequality, Polynomial Division, Absolute Value, Summation, Sequences, Simplify |
| Algebra | In-context | Add, Mul, Solve Equation, Sub, Div, Exp, Factoring, Quadratic Formula, Radicals, Distance Formula |
| | Internal | Absolute Value, Slope, Logarithm, Arithmetic Sequence, Completing the Square, Interval Notation, Inverse Function, Substitution, Midpoint Formula, Ceiling Function |
| Probability | In-context | Factorial, Combination, Counting Principle, Probability, Add, Sub, Permutations, Mul, Div, Exp |
| | Internal | Simplify Fraction, Binomial Theorem, Expected Value, Arithmetic Sequence, Sum of Arithmetic Series, Counting, Stars and Bars, Divisibility Rules, Binomial Probability, Perfect Squares |
| Pre-Calculus | In-context | Solve Equation, Add, Mul, Sub, Complex Number, Div, Factoring, Radicals, Area, Distance Formula |
| | Internal | Trigonometric Identities, Trigonometry, Dot Product, Matrix Multiplication, Pythagorean Theorem, Cross Product, Inverse Trigonometric Functions, Determinant, Vector Projection, Vectors |
| NumTheory | In-context | Add, Mod, Base Conversion, Mul, Congruences, Div, Sub, Factoring, Prime Number, GCD |
| | Internal | Divisors, Divisibility Rules, Units Digit, Prime Fraction, Chinese Remainder Theorem, Arithmetic Sequence, Exponents, Cyclic Patterns, Perfect Squares, Modular Arithmetic |

