# OpenReview forum: "Skills-in-Context Prompting:  Unlocking Compositionality in Large Language Models"
_ICLR.cc/2024/Conference — Submitted to ICLR 2024_

### Official Review · Reviewer_XFNL · 2023-10-31

**Soundness:** 3 good
**Presentation:** 1 poor
**Contribution:** 2 fair
**Rating:** 5
**Confidence:** 4

**Summary:**

This paper proposes a prompting technique called Skills-in-Context (SKiC) prompting. The motivation for this prompting method is that compositional generalisation is an ability that helps to solve complex problems that are composed of existing skills, and the prompting method explicitly decomposes tasks into sub-skills. The prompting method can be used manually, by having a human construct a prompt for a task, as well as semi-automatically, by prompting LLMs to construct the prompt and having a human verify it. The authors evaluate their prompting technique on a set of reasoning tasks and show it improves over existing prompting techniques, sometimes by a large margin. The authors also do an ablation study where they find that for the semi-automatic method it helps if the prompt is constructed by the same model that is being tested, and an error analysis to find the most common source of errors is required skills for a task that are not in the prompt.

**Strengths:**

The authors compare their method with a large set of baselines and convincingly show their method is better for the evaluated reasoning tasks.
The authors present the interesting finding that it helps for a model to generate its own prompt, even if another model might construct a prompt that is inherently more correct / high-quality.
The authors do an error-analysis and identify the most common source of errors.

**Weaknesses:**

This paper is written around the premise that this prompting method unlocks compositionality, but the way the authors use the term is unlike existing literature. Systematic compositionality is the ability to understand and produce novel combinations from known parts. This paper claims the prompting method SKiC *"teaches LLMs to generalize to harder problems than they have seen and to problems that require innovative compositions of existing knowledge (either in context or inside model weights)."* and that the authors *"develop an effective one-stage prompting strategy [..] to unlock the general compositional generalization capability in LLMs"*. However, the authors do not investigate at all whether the LLMs that are tested are trained on the tasks they give them. For example, the authors call the last-letter concatenation task of concatenating the last letter of 4 words or more *"out-of-distribution"*, but a simple Google search reveals an example for 5 words (https://tutorialspoint.com/print-last-character-of-each-word-in-a-string). It seems pretty likely to me that more examples are in the pre-training data of these models.
Furthermore, the nature of systematic generalisation requires near-perfect generalisation. For models that achieve less than 50% accuracy on some of the tasks, you cannot say they are generalising systematically, because they are not.
Other aspects of the paper make me think the authors have a different definition of systematic generalisation than most of literature, because they for example claim that Nye. et al (2021) develop the scratchpad method to *"unlock its potential compositional generalization capabilities"*, but there is nothing in that paper that indicates this is the case. Scratchpad prompting is developed to allow intermediate steps of computation and with it improves performance on multi-step computation problems.

To summarise; how do you know that LLMs have not been trained on the tasks you call "out-of-distribution"? Without knowing that, you cannot say the models are systematically generalising to unseen tasks. Additionally, when models do not reach near-perfect accuracy, it cannot be called systematic generalisation. I'd rephrame this paper to better reflect what it actually does, which is developing a prompting method that the authors empirically show to be superior to past prompting methods.

**Questions:**

How do you define "out-of-distribution"?

I encountered some typos etc. in the text:
	- "Towards this goal, there have been a series of prompting strategies being developed to improve the reasoning and compositionality capabilities." paragraph 2 introduction
	- "Put the asked words to a list. For example, put the words in 'apple' to D=['apple']"; Figure 1
	- "fine-tuneded baselines such as finetuned" page 6

---

> ### Author Response · Authors · 2023-11-21
>
> Thanks for your assessment and constructive feedback. We think that some concerns might be caused by misunderstanding, which we will explain in detail below.
>
>
> ## On whether LLMs are trained on the tasks and the the definition of ‘out-of-distribution’
> First, in this work, we follow the previous work [1,2,3] and focus on the easy-to-hard generalization. Specifically, the in-distribution/seen tasks here means the testing samples are sampled from the same problem size with what have been shown in the context [3]. For example, we demonstrate examples of 2-digit addition, and then test it over samples that are also from 2-digit addition. In contrast, the out-of-distribution/unseen tasks here are defined to be the harder unseen variants of the problem which are with different complexity compared to examples provided in the context.  For example, the testing samples of 5-digit additions are the harder variant of the problem that are not seen in the context examples (2-digit additions) [3]. And we propose the SKiC prompts to improve such easy-to-hard compositional generalization [3] and complex reasoning tasks compared to previous methods.  In some of the tasks, our SKiC actually achieves near-perfect performances on harder cases, which are not demonstrated in the context.
>
> In terms of the investigation on whether the LLMs have been pre-trained on some of the tasks, even if some of the tasks could be crawled into the pretraining corpus, we believe that they are mostly general and simple examples (e.g., last letters of 4 or 5 words)  rather than the harder cases that we tested on (e.g., last letters of 12 words). This is also demonstrated in the zero-shot performances on the harder cases: for example, the zero-shot performances of ChatGPT on last-letter, addition, multiplication and dynamic programming are quite low (lower than 50% in most of the cases)). With our SKiC prompting, the easy-to-hard generalization capability is significantly boosted to even near-perfect generalization, while other strong prompting methods such CoT and Least-to-Most cannot do so.
>
> [1]Decomposed Prompting: A MODULAR APPROACH FOR SOLVING COMPLEX TASKS
>
> [2]Least-to-Most Prompting Enables Complex Reasoning in Large Language Models
>
> [3] Faith and Fate: Limits of Transformers on Compositionality
>
>
>
> ## On the performances which are less than 50% accuracy
>
> On the tasks which do not reach near-perfect accuracy (i.e., multiplication), they are mainly constrained by the basic skills and the length of context windows. For example, our error analysis on the multiplication tasks  reveals that most of the errors in SKiC prompting are caused by missing the multi-digit addition capability, which can be incorporated as a basic skill in the prompting context.
>
> However, it would require an extra context window (over the limit when we perform the 4,5-digit multiplication) to calculate the addition following the adding SKiC prompts. So we did not include them which resulted in 48.5 performances with ChatGPT on 5-digits multiplication.
>
> When equipped with such extra skills, the most recent GPT-4 (128k context window) with SKiC can achieve near-perfect generalization. We further check the error cases and find that they are mainly caused by copying errors (copy wrong numbers from previous steps).
>
> | GPT4      | 2     | 3     | 4    | 5    |
> |-----------|-------|-------|------|------|
> | Zero-shot | 99.0  | 59.0  | 4.0  | 0    |
> | SKiC (without addition)      | 100.0 | 92.0 | 88.0 | 68.0 |
> | SKiC (with addition as basic skills)     | 100.0 | 100.0 | 99.0 | 93.0 |

---

> > ### Comment · Reviewer_XFNL · 2023-11-22
> > **thanks for the response**
> >
> > Thank you for your responses, however I remain with my original point that this paper does not address compositional generalisation because it doesn't discuss / investigate whether or not the tasks are part of the training data. Just because zero-shot generalisation is low doesn't mean that the problems aren't part of the training data, and if they are you cannot claim systematic generalisation. To be clear, I don't think this paper does not have merit or doesn't propose a strong method. As I said in my original review, it's a good work "developing a prompting method that the authors empirically show to be superior to past prompting methods." I just disagree with the current framing around compositionality, which is not what it tests, and hence should be re-written to reflect what it is actually about.

---

> > > ### Author Response · Authors · 2023-11-23
> > >
> > > Thank you for your feedback! We would like to add some further clarifications regarding “easy-to-hard/compositional generalization” and our contribution towards this direction.
> > >
> > > First of all, we kindly disagree that our work does not contribute to compositional generalization. Of course, we never claim that we have completely solved the compositional generalization problem. Instead, our contribution is developing a prompting strategy that can effectively boost the composition generalization performance. Compositional generalization refers to the ability to generate solutions to complex problems by recombining known foundational skills in a novel way. The tasks considered in this paper are all designed to measure the model capability in this regard, which were commonly used in related literature. Specifically, we follow previous work [1], where models need to use known skills (from in-distribution/easy problems) and generalize to unseen/complex/out-of-distribution problems. On these tasks, our proposed SKiC prompting can greatly boost the easy-to-hard performance compared to the previous state-of-the-art methods. For example, SKiC achieves near-perfect generalization on many tasks (e.g.,  last letters, addition, multiplication and dynamic programming) with huge improvement margin in the much harder testing cases, which was never observed before in earlier LLMs works. For tasks that are still not near-perfect, our improvement margin is still also huge in the regime of harder testing cases (for example, on the multihop QA task with ~15% improvements over [4]). Therefore, given the great performance improvement by SKiC over these compositional benchmarks, we believe our work should be regarded as a solid contribution towards improving easy-to-hard compositional generalization for LLMs.
> > >
> > > Second, we understand and value the reviewer’s feedback on examining the potential leakage of task data into the pretraining corpus of these LLMs. Unfortunately, the pretraining data for most of the LLMs are not publicly available even for the opensource models like LLAMA and LLAMA-2. We are also very eager to perform such analysis if the data are available. Due to such constraints, our best strategy to is try our best effort to ensure the harder testing cases (e.g., last letters of 12 words, 7 digits addition, 5 digits multiplication, finding max subsequence over 8 numbers) are not seen during pretraining. Generally, if a training data has been seen in LLMs, their zero-shot performance would be very high. And if the harder testing cases (e.g., 5 digits multiplication) exhibit zero performance, it is very unlikely that these testing cases have been seen in pretraining data. It might be possible that the easy versions of the problem (e.g., last letter of 2-3 words) are crawled into the pretraining corpus. However, it is very unlikely that the training process has seen the harder versions of the testing cases (e.g., concatenating last letters of 12 words, performing 5 digits multiplication, finding max subsequence over 8 numbers). In fact, compositional generalization allows  the model to see the easy versions of the examples (e.g., last letter concatenation of 2-3 words), as long as they have not seen the harder testing cases (e.g., last letter of 12 words). And all previous methods [1,4,5,6] have been shown to be struggling in solving these harder problems that require the composition of simple problems/basic skills, even though their easy variations might be seen in the pre-training stage. Compared to previous work, our method can drastically improve over that in the easy-to-hard testing cases. Further mitigate such a concern, we add a new experiment result where we test the last letter of 50 and 100 random strings with ChatGPT, which are reported below. It is very unlikely that such data are included in the pretraining data.
> > > | ChatGPT | Last Letter (50 words)  |Last Letter (100 words)  |
> > > |---------|-------|-------|
> > > | COT     | 62.0 |56.0|
> > > | SKiC    |100 |100|
> > >
> > >
> > >
> > >
> > >
> > >
> > > [1] Faith and Fate:Limits of Transformers on Compositionality
> > >
> > > [2] Compositional generalization in a deep seq2seq model by separating syntax and semantics
> > >
> > > [3] Good-Enough Compositional Data Augmentation
> > >
> > > [4] Decomposed Prompting: A MODULAR APPROACH FOR SOLVING COMPLEX TASKS
> > >
> > > [5] Least-to-Most Prompting Enables Complex Reasoning in Large Language Models
> > >
> > > [6] Chain-of-Thought Prompting Elicits Reasoning in Large Language Models

---

> > > ### Author Response · Authors · 2023-11-23
> > >
> > > Third, the tasks we utilized are consistently and widely used in previous works to show compositionality. For example, the multiplication and dynamic programming task [1]. Furthermore, we additionally conduct experiments on a dataset (SCAN) which is widely used to show compositional generalization [2,3,5].  And the performance with ChatGPT  is shown below (where our SKiC could achieve 100% accuracy). Note that, on the same foundation model, Chain-of-Thought (CoT) cannot achieve 100% generalization while our method can. Such a huge performance gain (+27.5%) is definitely a non-trivial contribution towards improving compositionality.
> > > | ChatGPT | SCAN  |
> > > |---------|-------|
> > > | COT     | 72.5  |
> > > | SKiC    | 100.0 |
> > >
> > > Finally, to further mitigate the reviewer’s concern, one alternative way to verify this is to come up with a brand new task that never existed before, which makes it impossible to be included in the pretraining corpus. We would greatly appreciate it if the reviewer could share some thoughts in this direction or recommend any such new tasks of this kind for evaluating the systematic generalization. We would be happy to immediately carry out the experiments and report the result here. This would definitely help us  investigate deeper into this problem.
> > >
> > >
> > >
> > >
> > > [1] Faith and Fate:Limits of Transformers on Compositionality
> > >
> > > [2] Compositional generalization in a deep seq2seq model by separating syntax and semantics
> > >
> > > [3] Good-Enough Compositional Data Augmentation
> > >
> > > [4] Decomposed Prompting: A MODULAR APPROACH FOR SOLVING COMPLEX TASKS
> > >
> > > [5] Least-to-Most Prompting Enables Complex Reasoning in Large Language Models
> > >
> > > [6] Chain-of-Thought Prompting Elicits Reasoning in Large Language Models

---

> > > > ### Comment · Reviewer_XFNL · 2023-11-23
> > > > **Thank you for the detailed response**
> > > >
> > > > I thank the authors again for the detailed responses, but I feel like they conflate two things: the contribution of this paper with a prompting method that can help with inherently compositional tasks, and the contribution of this paper with regards to compositional / systematic generalisation of large pre-trained transformers. The former you do not need to convince me, the paper does a good job.
> > > >
> > > > My problem even after these new results remains with the premise that with this methodology you can be sure that the models are using compositional generalisation. I agree with the authors that compositional generalisation refers to the application of known skills for novel problems. What I disagree with is that you can say with your experiments that the models are using compositional generalisation to generalise without knowing with some certainty that they have not been trained on the tasks.
> > > >
> > > > I agree with the authors that it's unlikely that a sequence of 100 random strings is part of the training data, but I disagree that zero-shot failure and SKiC success means that they are using compositional generalisation. There's many open questions about how these models generalise, and we do not know for sure that they cannot do something in-weights if they cannot do it zero-shot.
> > > >
> > > > I agree that it's difficult to test these kind of questions in the current LLM era where training data is either too large to analyse or not even publicly available, but that is not a reason to resort to methods for which it is unclear what type of generalisation the model is using. It has become an incredibly hard and interesting research problem to figure out what kind of generalisation technqiue these models are using, and I do not believe this prompting method convincingly enough shows that they are compositionally generalising to make the strong claims that are made in the paper. I remain firm believing this paper should be re-written to reflect what it actually does, which is achieve high performance on compositional tasks with a novel prompting technique over other known prompting techniques. Reviewing such rewrite, I would happily recommend acceptance. However, from the current experiments it's difficult to say if they are generalising compositionally or not, at best you can say they might be and it requires further study to be sure.

---

### Official Review · Reviewer_52w8 · 2023-11-01

**Soundness:** 3 good
**Presentation:** 3 good
**Contribution:** 2 fair
**Rating:** 5
**Confidence:** 3

**Summary:**

This paper proposes a prompting approach to improve compositional generalization in large LMs through an in-context learning approach. The approach is as follows:

- First, provide the model with "skills" along with some examples showing how to invoke skills
- Then, provide some examples of how to compose these skills to solve a task
- Then with these demonstrations, see if the model can generalize to solving tasks that involve new compositions of tasks.

As with all prompting approaches, the main question is how much human knowledge / time went into designing the prompt and how generally applicable is the approach. Having said that, from results we find that the proposed approach (SKIC for Skill in-context prompting) improves performance over zero-shot and CoT prompted LMs, on challenging tasks including long digit addition / multiplication

**Strengths:**

- Lots of experiments of a simple approach, and very good results across the board.
- The approach seems straightforward and easy to apply.
- One of the novel aspects in the paper was to use an LM to mine a set of primitive skills from a dataset of demonstrations and then use the same LM to compose these skills. Here, there is much lower human intervention (though there are a few details I couldn't quite get about which experiments use author provided skills vs LM generated skills).

**Weaknesses:**

I think this work could have explored some of the technical aspects of "compositional" skill utilization in LMs beyond simple prompting:

1. What are the limitations of this kind of generalization? Are there some compositions that all models fail at? How do you reconcile this apparent success at compositional generalization with other work that shows poor compositionality in LMs?
2. Can this be used for semantic parsing tasks where an LM is prompted to discover skills?
3. Can this be applied to general NLU tasks (e.g. textual entailment) by prompting an LM to discover skills?
4. Can the LM discover the somewhat "hand-designed" skills of "extracting digits" / "list length" for addition? If not, what skills does the LM discover there?

Currently, the paper lacks answers to a lot of these technical questions. It's also not clear if this approach is generally applicable. The notion of a skill may not always be very clearly described in terms of natural language. But i'm happy to improve my score if some of these weaknesses are addressed.

**Questions:**

- There is a whole sub-field in language and robotics where LLMs are used to decompose plans into sub-tasks. How do those approaches compare to SkiC?

---

> ### Author Response · Authors · 2023-11-21
>
> Thank you for the insightful feedback. Hope our clarification below addresses your comments.
>
> ## On the limitations of easy-to-hard generalization
>
> Previous work [1,2,3] introduced step-by-step reasoning and breaking down hard problems to simple problems to improve the easy-to-hard generalization. However, in our work, we make another important discovery that, in order to teach models how to compose skills, it is also crucial to demonstrate the foundational skills and how to ground each of its reasoning steps onto the foundation skills. That is, besides step-by-step reasoning, explicit grounding is another key factor to elicit compositionality and easy-to-hard generalization. We achieve that by proposing SKiC and our SKiC shows significantly better performances compared to previous work in all the experiments. Additionally, we add extra ablation study to highlight our finding (the importance of skill grounding in reasoning steps). We compare SKiC with the setting where (i) we remove the skills but keep the skill grounding in reasoning steps and (ii) we remove the skill grounding in reasoning steps but keep the basic skill introduction in the front. The performance on Dynamic Programming is shown below. Removing both parts would bring in the performance drop, which further indicates the importance of skills and skill grounding in reasoning steps to improve the easy-to-hard generalization.
>
> | ChatGPT | DP (8 numbers) |
> |-------------------|----------------|
> | COT | 72.0 |
> | SKiC | 98.0 |
> | - skills | 94.0 |
> | - skill grounding | 82.0 |
>
>
>
> [1]Decomposed Prompting: A MODULAR APPROACH FOR SOLVING COMPLEX TASKS
>
> [2]Least-to-Most Prompting Enables Complex Reasoning in Large Language Models
>
> [3]  Faith and Fate: Limits of Transformers on Compositionality

---

> > ### Author Response · Authors · 2023-11-21
> >
> > ## Can this be used for semantic parsing tasks?
> >
> > Yes, our SKiC could be easily used for semantic parsing tasks. For example, we further design prompts and perform experiments on SCAN dataset [1]. Specifically our skills include <decompose_commands> and <command_parse>:
> >
> > ```
> >
> > Skill <command_parse>: Parse a simple command into structured output.
> >
> > For example,
> >
> > ‘Walk’ is parsed to ‘WALK’
> >
> > ‘Look’ is parsed to ‘LOOK’
> >
> > ‘Run’ is parsed to ‘RUN’
> >
> > ‘Jump’ is parsed to ‘JUMP’’
> >
> > ‘turn left’ is parsed to ‘LTURN’’
> >
> > ‘turn right’ is parsed to ‘RTURN’’
> >
> > ‘Turn opposite left’ is parsed to ‘LTURN LTURN’
> >
> > ‘Turn opposite right’ is parsed to ‘RTURN RTURN
> >
> >
> >
> > Skill <decompose_commands>: decompose the complex commands into basic units.
> >
> > For example,
> >
> > ‘Look right’ can be decomposed to ‘look’ and ‘turn right’
> >
> > ‘Look twice’ can be decomposed to ‘look’ and ‘look’
> >
> > ‘Look right after look twice’ can be decompose to ‘look right’ and ‘look twice’
> >
> > ‘Jump opposite right thrice and walk’ can be decomposed to ‘jump opposite right thrice’, ‘walk’
> >
> > ```
> >
> > One example of composing the above skills is also shown below
> >
> > ```
> >
> > Example: run and look twice.
> >
> > Answer: 1. Using the Skill <decompose_commands>, ‘run and look twice’ can be decomposed to ‘run’, and ‘look twice’.
> >
> > 2. For ‘run’, using the Skill <command_parse>, ‘run’ is parsed to ‘RUN’
> >
> > 3. For ‘look twice’, using the Skill <decompose_commands>, ‘look twice’ can be decomposed to ‘look’ and ‘look’
> >
> >       i. For ‘look’, using the Skill <command_parse>, ‘look’ is parsed to ‘LOOK’
> >
> >       ii. For ‘look’, using the Skill <command_parse>, ‘look’ is parsed to ‘LOOK’
> >
> > 4. The output is ‘RUN’+’LOOK’+’LOOK’ = ‘RUN’ + ‘LOOK’*2
> >
> > ```
> >
> > And the performance with ChatGPT is shown below:
> >
> > | ChatGPT | SCAN |
> > |---------|-------|
> > | COT | 72.5 |
> > | SKiC | 100.0 |
> >
> >
> >
> > [1] Compositional generalization via neural-symbolic stack machines

---

> > > ### Author Response · Authors · 2023-11-21
> > >
> > > ## Can LM discover skills and be applied to general NLU tasks textual entailment?
> > >
> > > Yes, LM can also be prompted to discover skills for general tasks.
> > >
> > > First, in our second setting, we have shown that the skills can be directly distilled from LLM by prompting ‘what are some basic skills in algebra’, which derive skills shown in the Figure 17.
> > >
> > > Additionally, we further provide experiments to show that some of the skills in our first setting can also be summarized by LLMs and our SKiC could also be applied to general NLU settings like NLI tasks.
> > >
> > > ### Discover skills for last letters.
> > > We provide ChatGPT with the chain-of-thought prompts for last-letter tasks and ask it to summarize basic skill to solve the tasks:
> > >
> > > ```
> > > Summarize the basic skills to solve last-letter tasks from the examples below:
> > >
> > >
> > > Q: “think, machine”
> > >
> > > A: The last letter of “think” is “k”. The last letter of “machine” is “e”. Concatenating “k”, “e” leads to “ke”. So, “think, machine” outputs “ke”.
> > >
> > > Q: “learning, reasoning, generalization”
> > >
> > > A: The last letter of “learning” is “g”. The last letter of “reasoning” is “g”. The last letter of “generalization” is “n”. Concatenating “g”, “g”, “n” leads to “ggn”. So, “learning, reasoning, generalization” outputs “ggn”.
> > >
> > > ```
> > >
> > > The LLMs (ChatGPT) would further summarize the needed basic skills:
> > >
> > > ```
> > >
> > > To solve last-letter tasks based on the given examples:
> > >
> > > -   Identify Words: Recognize the words provided in the task.
> > >
> > > -   Determine Last Letters: Find the last letter of each identified word.
> > >
> > > -   Concatenate Last Letters: Combine the last letters in the specified order.
> > >
> > > -   Form New Sequence: The result is a new sequence created by concatenating the last letters.
> > >
> > >
> > > In the provided examples:
> > >
> > > -   For "think, machine": Identify "k" (last letter of "think") and "e" (last letter of "machine"), then concatenate to form "ke."
> > >
> > > -   For "learning, reasoning, generalization": Identify "g" (last letter of "learning"), "g" (last letter of "reasoning"), and "n" (last letter of "generalization"), then concatenate to form "ggn."
> > >
> > >
> > > ```
> > >
> > > The discovered skills include ‘Identify Words’, ‘Determine Last Letters’, ‘Concatenate Last Letters’, ‘Form New Sequence’, which are similar to what we have shown in Figure 6. With such skills, we could further construct the SKiC prompts by adding these basic skills in the context and grounding reasoning steps onto these basic skills. This gives the similar performance compared to what we constructed manually.
> > >
> > > | ChatGPT | Last Letter (12) |
> > > |---------------------------------|------------------|
> > > | COT | 72.5 |
> > > | SKiC | 100.0 |
> > > | SKiC (Skills discovered by LLM) | 100.0 |

---

> ### Author Response · Authors · 2023-11-21
>
> ### Discover skills for NLI (we use RTE as an example).
> We first randomly sample two examples from RTE training dataset and generate the predictions.
>
> ```
>
> Q: Please identify whether the premise entails the hypothesis.
> premise: Dana Reeve, the widow of the actor Christopher Reeve, has died of lung cancer at age 44, according to the Christopher Reeve Foundation.
> hypothesis: Christopher Reeve had an accident
>
> ChatGPT: The premise and hypothesis are separate statements, and the premise does not entail the hypothesis. The premise provides information about Dana Reeve, the widow of Christopher Reeve, and her cause of death, which is lung cancer. The hypothesis, on the other hand, is about Christopher Reeve having an accident, and it is not directly related to the information provided in the premise.
>
>
> Q: Please identify whether the premise entails the hypothesis.
> premise: Security forces were on high alert after an election campaign in which more than 1,000 people, including seven election candidates, have been killed.
> hypothesis: Security forces were on high alert after a campaign marred by violence
>
> ChatGPT: Yes, the hypothesis is consistent with the premise. The premise states that security forces were on high alert after an election campaign in which more than 1,000 people, including seven election candidates, have been killed. The hypothesis aligns with this by stating that security forces were on high alert after a campaign marred by violence, which corresponds to the information provided in the premise.
>
> ```
>
> Then we prompt ChatGPT to discover the skills:
>
> ```
> Discover the skills from above examples to do the NLI tasks:
>
> ```
>
> Which result in the following output:
>
> ```
>
> The basic skills to solve NLI (Natural Language Inference) tasks involve understanding the relationship between the premise and hypothesis statements. Here are the key skills demonstrated in the examples:
>
> -   Context Understanding: Understanding the context and contextually relevant information is key. In both examples, comprehending the context provided in the premises is essential to evaluate the relationship with the hypotheses. This includes understanding details about individuals, events, and the cause-and-effect relationships presented in the text.
>
> -   Inference Evaluation: The skill to evaluate whether the hypothesis is supported, contradicted, or unrelated to the information in the premise. In the first example, the inference evaluation involves determining that the premise about Dana Reeve's death does not support the hypothesis about Christopher Reeve's accident. In the second example, it involves recognizing that the premise supports the hypothesis about security forces being on high alert due to campaign violence.
>
>
> These skills collectively enable a reader or a model to make accurate assessments of the logical connections between statements in NLI tasks.
>
>
>
> ```
>
> We could summarize basic skills from the LLMs: Context Understanding, Inference Evaluation. And we can further form the SKiC prompts:
>
> ```
>
> Skill <Context Understanding>: Understanding the context and contextually relevant information.
>
> For example,
>
> ‘Dana Reeve, the widow of the actor Christopher Reeve, has died of lung cancer at age 44, according to the Christopher Reeve Foundation.’
>
> The sentence provides information about Dana Reeve, the widow of Christopher Reeve, and her cause of death, which is lung cancer.
>
> ‘Christopher Reeve had an accident’
>
> The sentence provides information about Christopher Reeve having an accident
>
>
>
> Skill <Inference Evaluation> The skill to evaluate whether the hypothesis is supported, contradicted, or unrelated to the information.
>
> For example,
>
> Premise: Dana Reeve
>
> Hypothesis: Christopher Reeve having an accident
>
> Using the skill <Inference Evaluation>, the hypothesis is not related to premise.
>
>
>
>
> Compose the above skills to solve the NLI tasks.
>
> Example #1:
>
> premise: Dana Reeve, the widow of the actor Christopher Reeve, has died of lung cancer at age 44, according to the Christopher Reeve Foundation.
>
> hypothesis: Christopher Reeve had an accident
>
> Answer: 1. Using the skill <Context Understanding>, The premise provides information about Dana Reeve, the widow of Christopher Reeve, and her cause of death, which is lung cancer.
>
> 2. Using the skill <Context Understanding>, the hypothesis is about Christopher Reeve having an accident.
>
> 3.  Using the skill <Inference Evaluation>, the information in hypothesis is not directly related to the information provided in the premise. So the answer is Not entail.
>
>
> ```
>
>
>
> The SKiC prompts constructed by utilizing the skills discovered from LLMs could achieve better performances on RTE compared to COT baselines.
>
> | ChatGPT | RTE |
> |---------------------------------|------|
> | COT | 85.2 |
> | SKiC (Skills discovered by LLM) | 89.8 |
>
> With the extra experiments in semantic parsing and discovering skills for lastletter and NLI, we believe that SKiC is generally applicable.

---

> > ### Author Response · Authors · 2023-11-21
> >
> > ## On the relations with language and robotics field
> >
> > We think our SKiC can be easily adapted to the research in language and robotics domains where the ability to decompose plans can be viewed as a specific skill similar to what we have done for multi-hop QA tasks in Table 3 and each specific sub-tasks can be solved by basic skills specified in the context (constraints on the action space of robots).

---

### Official Review · Reviewer_v2fa · 2023-11-01

**Soundness:** 2 fair
**Presentation:** 3 good
**Contribution:** 2 fair
**Rating:** 3
**Confidence:** 5

**Summary:**

This paper proposes a prompting method termed as "SKiC" (skills-in-context), which equips the large language models (LLMs) with a set of pre-defined skills by leveraging their in-context learning capabilities. First of all, the method constructs a set of potential skills either from human annotation (distill skills via human) or model generation (distill skills via prompting LLMs). The authors find it is crucial to demonstrate both the skills and the compositional examples within the SKiC prompts. Also, SKiC prompting is shown to be able to solve unseen complex reasoning problems better than previous methods (CoT/Decomp/etc.).

**Strengths:**

**Clarity & Significance**

- The effectiveness of the proposed method is clearly demonstrated through extensive experiments. The empirical observations indicate that the proposed method indeed improves the performance of LLMs across a range of sizes (davinci-003, ChatGPT, GPT4, etc.). The improvement is significant in many tasks, compared to several prior arts.
- The overall presentation of this paper is clear and easy to follow. The evaluation including ablation studies is comprehensive. The appendix is very detailed.

**Weaknesses:**

**Novelty**

I believe the idea of including skills in context has been proposed in many previous works, which made me unsure about its novelty and whether the contribution of this work is significant. For example, [1] also focuses on the challenge of large number addition/multiplication, and the authors also proposed to include basic skills and their composition in the context. They have also shown significant improvement with in-context skills prompting when there are more digits involved. I'd love to hear from the authors what the novelty of this work is given [1] was proposed and publicly available nearly one year ago.

However, many recent works have found using tools can significantly improve performance in math reasoning tasks (e.g., GSM8k and MATH), such as PAL [2] and Program-of-Thought [3], and they have shown that tool-using skills (rather than natural or symbolic language skills) can bring more gain to math reasoning tasks. I understand the main focus of this work is not about tool-using, but the method proposed here is a bit less appealing to me considering the current SoTA can reach 80+ (see [4]), especially when considering [1] already discussed the tool-using case.

**Soundness**

My main concern is that the skills used in the SKiC are somehow mined from the test set (e.g., you create basic addition/multiplication skills and the compositional examples for GSM8k because you pre-know it is a math reasoning task and preview some of its questions), which might not be the case in practice. This raises the question of how the framework's performance would be affected if the skills were taken from a different set of tasks. For example, will I see any improvement in BBH object counting or penguins on a table task (both are about numeric reasoning) with the SKiC skills you designed for GSM8k? In other words, I guess there is a pre-condition of the success of SKiC, which you can somehow preview the test set of the target tasks, and then you can prepare the needed skills beforehand (either from human annotation or from LLM prompting), which I don't believe is the true definition of "solving unseen tasks".

While the paper does not explicitly address this issue, it is possible that the performance of the framework could be affected if the skills were taken from a different set of tasks. This is because the skills used in the framework are specific to the domain of the target task test set, and may not be applicable to other domains. For example, if the skills were taken from the Last Letter Concatenation and tested on the GSM8K/MATH tasks, I don't think SKiC can lead to optimal performance.

[1] Teaching Algorithmic Reasoning via In-context Learning

[2] PAL: Program-aided Language Models

[3] Program of Thoughts Prompting: Disentangling Computation from Reasoning for Numerical Reasoning Tasks

[4] Solving challenging math word problems using gpt-4 code interpreter with code-based self-verification

**Questions:**

**Impact of the in-context compositional examples**

1. How did you pick the two/N examples in your prompts? I believe the performance will be heavily affected by the choice of these examples. Did you run experiments or have a discussion on their impact?
2. Does the order of these examples matter? Telling from [1] I think the order is another factor that will affect the performance, but the work does not discuss it.


[1] Calibrate Before Use: Improving Few-Shot Performance of Language Models

---

> ### Author Response · Authors · 2023-11-21
>
> Thanks for your insightful comments. We humbly think that some concerns might be caused by misunderstanding, which we will clarify in detail below. We hope that our response can clarify the misunderstandings and you can consider our work more favorably.
>
> ## On the difference from previous work “Teaching Algorithmic Reasoning via In-context Learning”
>
> The major goal in [1] is to design algorithm-style prompts to first teach basic math skills like adding two numbers and then decompose hard problems into linear subproblems which could be solved by the usage of previously taught math operations such as multiplication through adding. This is actually similar to [2] which has been discussed in our Section 2.1.
>
> Our work is different and better than [1] in several aspects:
>
> (i) First, beyond algorithmic operation, we emphasize the explicit grounding of general basic skills and teaching models to compose multiple basic and grounded skills to improve the easy-to-hard generalization and the performances on complex tasks. For example, our SKiC for adding two number teach models to compose extract_digits skills, list_length skills, add_two_single_digit_number skills, sub_10 skills and compare_10 skills compare to the algorithmic style prompt in [1] and achieve better performances on adding tasks with ChatGPT as shown below:
>
> | Add | 2 | 3 | 4 | 5 | 6 | 7 |
> |---------------|-----|-----|------|------|------|------|
> | COT | 100 | 100 | 98.5 | 90.0 | 87.5 | 80.0 |
> | Algorithm [1] | 100 | 100 | 98.0 | 94.5 | 91.5 | 90.0 |
> | SKiC | 100 | 100 | 99.0 | 98.0 | 99.0 | 98.5 |
>
>
>
> (ii) Second, not all the problems can be decomposed linearly and follow [1] to solve. Even though [1] formalize multiplication as a composition of multiple adds (e,g, 11*3 = (11+11) + 11), it is impossible and slow to call multiple adds when it comes to larger numbers like 128*128 (doing 128 additions is infeasible in terms of error propagation and computation cost (significantly longer context window or significant amount of extra calls of basic skills). Instead, our SKiC encourages models to perform multi-digits multiplication by composing single-digit multiplication skill and adding skill in only one stage, which also achieves significantly better performance with one single calls of the foundation model (e.g., ChatGPT) as shown below (when dealing with multiplication between two 3/4/5 digits number, linearly decomposition like [1] would result in exceeding context length.):
>
> | Mul | 2 | 3 | 4 | 5 |
> |---------------|-----|------|------|------|
> | COT | 99 | 54.5 | 13.0 | 2.0 |
> | Algorithm [1] | 100 | 68.0 | 20.0 | 0 |
> | SKiC | 100 | 82.0 | 72.0 | 48.5 |
>
> (iii) Third, our SKiC also encourages LLMs to utilize their internal skills beyond what have been specified in the context to solve complex math reasoning tasks like GSM8K and MATH, which can not be achieved by [1] which are limited to the usage of provided skills. Our proposed SKiC actually enables LLMs to utilize a wider range of internal skills to solve problems as shown in Figure 27, 28, 29, 30, 31, 32, 33 via automatically generating skills from LLM itself and grounding to both mentioned skills and unmentioned skills in examples. The overall performance on GSM8K with ChatGPT also demonstrated the powerfulness of our SKiC compared to algorithm calls in [1].
>
> | ChatGPT | COT | Algo [1] | SKiC |
> |---------|------|----------|------|
> | GSM8K-addition-only SubSet | 78.8 | 84.8 | 90.5 |
> | GSM8K | 74.4 | 79.5 | 87.2 |
>
>
>
>
>
> [1] Teaching Algorithmic Reasoning via In-context Learning
>
> [2] Decomposed Prompting: A MODULAR APPROACH FOR SOLVING COMPLEX TASKS

---

> > ### Author Response · Authors · 2023-11-21
> >
> > ## On the comparison to tool-using work
> >
> > The major contribution of our work is to understand and unlock the inherent composition abilities (easy-to-hard generalization) in LLMs themselves. The line of tool-using work is actually complementary with our proposed work and can be easily integrated to substitute several basic skills to further improve the performances.
> >
> > First, external tools like programs might bring in extra latency during inferences as LLMs need to call multiple external functions when dealing with complex problems. As a result, we mainly focus on whether LLM itself could solve tasks in one-stage through our SKiC.
> >
> > Second, it is hard/impossible to enumerate all the needed external skills (external calls) in the context for complex tasks, which would lower down the generalization abilities if the models are taught to rely on provided external calls. So, our SKiC also encourages models to utilize their internal skills which are not provided in the context to solve complex tasks.
> >
> > What’s more, tool-using cases are more focused on math-related reasonings. Not all the tasks can be improved by external tools (e.g., QA in our Table 3). Our SKiC is more general to different types of tasks, and the tool-use is actually one basic skill that could be integrated into SKiC. We leave this as a future work.
> >
> > In terms for the performance on the MATH dataset, 80+ achieved in [1] are mainly due to the usage of GPT4-code-interpreter (a stronger backbone model api which could achieve 69.69 by itself compared to 42.20 from GPT4 and 31.1 from ChatGPT which are our backbone model) and majority voting (majority voting over 16, 73.54 without majority voting). We utilize the code-based-verification in [1] without majority voting on MATH with ChatGPT and it actually achieves worse performances compared to our SKiC:
> >
> > | ChatGPT | COT | CSV [1] | SKiC |
> > |---------|------|---------|------|
> > | MATH | 31.1 | 34.7 | 40.6 |
> >
> > [1] Solving challenging math word problems using gpt-4 code interpreter with code-based self-verification

---

> > > ### Author Response · Authors · 2023-11-21
> > >
> > > ## On the designed skills
> > >
> > > In our work, we come up with two ways to construct the skills in SKiC. And in both ways, we are not using any of the testing data to mine the skills: (i) In the first way where we construct the skill set manually, for last letters, addition, multiplication, dynamic programming, we randomly generating 2 cases and summarize the potential skills, and for QA and GSM8K, we randomly sample 4 cases from their training set to summarize the skills. (ii) In the second way where we prompt LLMs to generate skills, we did not utilize any training or testing samples. We only prompt the model to generate the skills (e.g. prompting the LLM with “what are the basic skills in algebra?”).
> > >
> > > Dating back to GPT2 and GPT3, the knowledge/preview of the target tasks is also the essential component to design few-shot prompts in all the previous work [1,2,3,4,5], where all the examples in designed prompts are actually task-specific. Additionally, the number of examples we used are also limited (2 to 4 examples).
> > >
> > > We still observe performances improvements even if the skills in the context are taken from different tasks because our SKiC encourage the models to utilize and compose skills beyond what are included in the context: We use the skills and prompts designed for GSM8K and directly apply them on MATH (math reasoning), FOLIO (logical inference) which are unseen tasks as shown below.
> > >
> > > | ChatGPT | COT for GSM8K | SKiC for GSM8K |
> > > |---------|---------------|----------------|
> > > | MATH | 28.2 | 31.34 |
> > > | FOLIO | 68.8 | 72.5 |
> > >
> > >
> > >
> > > [1] Teaching Algorithmic Reasoning via In-context Learning
> > >
> > > [2] Decomposed Prompting: A MODULAR APPROACH FOR SOLVING COMPLEX TASKS
> > >
> > > [3] Solving challenging math word problems using gpt-4 code interpreter with code-based self-verification
> > >
> > > [4] Least-to-Most Prompting Enables Complex Reasoning in Large Language Models
> > >
> > > [5] Complexity-based prompting for multi-step reasoning

---

> > > > ### Author Response · Authors · 2023-11-21
> > > >
> > > > ## On the definition of ‘solving unseen tasks’
> > > >
> > > > To make clarifications, the ‘solving unseen tasks’ in our work refers to easy-to-hard generalization following the convention used in previous works [1,2,3], where models see the simple examples but need to generalize to harder examples for the same tasks (e.g., seeing examples of 2-digits adding but inferring how to add 5 digits numbers).
> > > >
> > > > However, in our extra experiments, we also show that our SKiC which teach the model how to compose skills can also help the performances even if the provided prompts are designed for different tasks: We use the skills and prompts designed for GSM8K and directly apply them on MATH (math reasoning), FOLIO (logical inference) which are unseen tasks as shown below.
> > > >
> > > > | ChatGPT | COT for GSM8K | SKiC for GSM8K |
> > > > |---------|---------------|----------------|
> > > > | MATH | 28.2 | 31.34 |
> > > > | FOLIO | 68.8 | 72.5 |
> > > >
> > > >
> > > >
> > > >
> > > >
> > > > [1]Decomposed Prompting: A MODULAR APPROACH FOR SOLVING COMPLEX TASKS
> > > >
> > > > [2]Least-to-Most Prompting Enables Complex Reasoning in Large Language Models
> > > >
> > > > [3]  Faith and Fate: Limits of Transformers on Compositionality

---

> > > > > ### Author Response · Authors · 2023-11-21
> > > > >
> > > > > ## On the questions about how to pick two/N examples in prompts
> > > > >
> > > > > We randomly pick the examples. For instance, the examples for the last-letter task are randomly sampled (we randomly set the words in the example like the last letter of ‘apple, pie’). The performance improvements are consistent even if we perturb the examples in the prompts. The results on last-letter tasks with ChatGPT with the use of different choices of few-shot examplars in the prompts are shown below. It shows the robustness of our proposed SKiC prompt to the selection of the few-shot exemplars.
> > > > >
> > > > > | ChatGPT | COT | SKiC |
> > > > > |-----------------------------------|------|-------|
> > > > > | 'apple, banana'; 'apple, pie' | 91.4 | 100.0 |
> > > > > | 'math, code'; 'science, computer' | 92.5 | 100.0 |
> > > > > | 'ashc, edhoh'; 'shbod, wojois' | 90.8 | 100.0 |
> > > > >
> > > > >
> > > > >
> > > > > ## On the order of the examples
> > > > >
> > > > >
> > > > >
> > > > > The order of the examples also does not matter a lot because we randomly sample a limited number of examples (2 examples in most of the cases) to design SKiC. We shuffle the order in our prompts (consisting of 4 examples) for GSM8K and the performances are shown below:
> > > > >
> > > > >
> > > > >
> > > > > | ChatGPT | COT | SKiC |
> > > > > |---------|------|------|
> > > > > | order 1 | 74.4 | 87.2 |
> > > > > | order 2 | 73.8 | 86.9 |
> > > > > | order 3 | 73.0 | 87.8 |

---

> ### Author Response · Authors · 2023-11-21
> **One example of the prompt and output for FOLIO task when using our SKiC prompts designed for GSM8K task**
>
> One example of the prompt and output for FOLIO task when using our SKiC prompts designed for GSM8K task is
> ```
> >>> prompt
> [SKiC: skills for GSM8K]
> [SKiC: examples for GSM8K]
> Following the above examples, use your skills (not limited to the skills provided above) to solve the logical inference problem: Given a set of premise, decide whether the conclusion can be derived from the premise.
>
> Premise: ["All people who regularly drink coffee are dependent on caffeine.", "People either regularly drink coffee or joke about being addicted to caffeine.", "No one who jokes about being addicted to caffeine is unaware that caffeine is a drug.", "Rina is either a student and unaware that caffeine is a drug, or neither a student nor unaware that caffeine is a drug.", "If Rina is not a person dependent on caffeine and a student, then Rina is either a person dependent on caffeine and a student, or neither a person dependent on caffeine nor a student. "]
> Conclusion: "Rina is a person who jokes about being addicted to caffeine or unaware that caffeine is a drug."
> Use all your skills (not limited to the skills provided above)  to check: Can the conclusion be drawn based on the premise (answer True, False or Unknown)?
> Answer:
>
> >>> ChatGPT output:
> <C>
> 1. People either regularly drink coffee or joke about being addicted to caffeine. Using the Skill <Or>, this can be expressed as: "People who do not regularly drink coffee either joke about being addicted to caffeine or are unaware that caffeine is a drug."
> 2. No one who jokes about being addicted to caffeine is unaware that caffeine is a drug. Using the Skill <Not>, this can be expressed as: "People who joke about being addicted to caffeine are aware that caffeine is a drug."
> 3. Rina is either a student and unaware that caffeine is a drug, or neither a student nor unaware that caffeine is a drug. Using the Skill <Or>, this can be expressed as: "Rina is either a student and unaware that caffeine is a drug, or Rina is neither a student nor unaware that caffeine is a drug."
> 4. If Rina is not a person dependent on caffeine and a student, then Rina is either a person dependent on caffeine and a student, or neither a person dependent on caffeine nor a student. Using the Skill <If-Then>, this can be expressed as: "If Rina is not a person dependent on caffeine and a student, then Rina is either a person dependent on caffeine and a student, or neither a person dependent on caffeine nor a student."
> 5. All people who regularly drink coffee are dependent on caffeine. Using the Skill <All>, this can be expressed as: "People who regularly drink coffee are dependent on caffeine."
> 6. Combining the information, we can infer that:
>     i. Rina is either a person dependent on caffeine and a student, or neither a person dependent on caffeine nor a student (from premise 4).
>     ii. If Rina is a person dependent on caffeine and a student, then she is aware that caffeine is a drug (from premise 1).
>     iii. If Rina is neither a person dependent on caffeine nor a student, then she is neither aware that caffeine is a drug nor jokes about being addicted to caffeine (from premise 3).
>     iv. If Rina is not a person dependent on caffeine and a student, then Rina is either a person dependent on caffeine and a student or neither a person dependent on caffeine nor a student (from premise 4).
>     v. If Rina is a person dependent on caffeine and a student, then she jokes about being addicted to caffeine (from premise 2).
>     vi. If Rina is neither a person dependent on caffeine nor a student, then she does not joke about being addicted to caffeine (from premise 3).
> 7. Combining the information from steps 6 and 5, we can further infer that:
>     i. If Rina is a person dependent on caffeine and a student, then she jokes about being addicted to caffeine.
>     ii. If Rina is neither a person dependent on caffeine nor a student, then she does not joke about being addicted to caffeine.
> 8. Therefore, the conclusion "Rina is a person who jokes about being addicted to caffeine or unaware that caffeine is a drug" can be drawn based on the premise.
> 9. The answer is True.
> </C>
> ### True
> <EOD>
> ```
>
> Note that instead of using the skills we provided in the context for GSM8K, the model is actually learnt to use their internal skills like Skill <Or> for this logical inference task.

---

### Official Review · Reviewer_wLQH · 2023-11-01

**Soundness:** 4 excellent
**Presentation:** 4 excellent
**Contribution:** 4 excellent
**Rating:** 8
**Confidence:** 4

**Summary:**

This paper proposes a new prompting approach for compositional reasoning, which forms exemplars as skills which can be flexibly composed by the LLM. This guides the LLM to ground its reasoning steps on skills which are already available in its knowledge. The method significantly improves performance on compositional generalization, achieving the  state-of-the art.

**Strengths:**

The idea proposed in the paper is intuitive but effective, and the paper is very well written.
The performance of the model is convincing, and the results show impressive performance on both composition over in-context skills and beyond in-context skills.

**Weaknesses:**

Tables 1 and 2 could be better organized in the page.

**Questions:**

1. In the beyond in-context skills setting, does the LLM use skills that are not specified in Figure 16?
2. Is it possible to replace some of these skills with function calls to Python, for example, to decrease computation errors?
3. More broadly what would be the pros and cons of letting the model access such "external" skills (if it is possible)?

---

> ### Author Response · Authors · 2023-11-21
>
> Thanks for your positive assessment and constructive feedback.
>
>
>
> ## On the question about the use of skills in the beyond in-context skills setting
>
>
>
> Yes, in the beyond in-context skills setting, our methods would encourage models to use skills in their pre-trained knowledge that are not provided in the provided context.
>
>
>
> For GSM8K, Figures 27 and 28 show the cases where models utilize skills that are not specified in Figure 16 (e.g., “round” and “compare”)
>
>
>
> For MATH, Figure 29, 30, 31, 32, 33 show the examples where models utilize skills that are not specified in context (Figure 17). Also, Table 12 in the Appendix shows the most frequent skills that are from Figure 16 and the most frequent skills that are not specified in Figure 17.
>
>
>
> ## The possibility to replace some of the skills with function calls
>
>
>
> Yes, it is possible to replace several skills with function calls like python which could be used to decrease some computation errors. This is complementary to our proposed methods. But in this work, the major goal is to explore the inherent capabilities of the language model itself. That is, we would like to examine and unlock the inherent abilities of LLMs to compose different skills for complex tasks.
>
>
>
> ## The pros and cons of letting the model access ‘external’ skills
>
>
>
> The pros of using ‘external’ skills (such as functional / api calls) is that we can even further boost performance by incorporating additional strong basic skills. For example, if we provide an api call of python interpreter as an external basic skill, it can generally reduce the naive computation error involved in math reasoning tasks.
>
>
>
> The cons are:
> (i) External calls might bring in extra latency during inferences as LLMs need to call external skills multiple times in multiple stages. As a result, we mainly focus on whether LLM itself could solve tasks in one-stage through our SKiC.
> (ii) It is hard/impossible to enumerate all the needed external skills (external calls) in the context for complex tasks, which would lower down the generalization abilities if the models are taught to rely on provided external calls. So, our SKiC also encourages models to utilize their internal skills which are not provided in the context to solve complex tasks.
>
>
>
> In a nutshell, the main focus of this paper is to investigate the inherent capability of compositional generalization by LLM itself. Integrating with external functional calls as additional basic skills is complementary to this work, and we leave it as a future work.
>
>
>
> ### The organization of Table 1 and 2
>
> We will better distribute Table 1 and 2 in the revised version.

---

### Author Response · Authors · 2023-11-21

We want to thank all the reviewers for their assessment and insightful comments. Per reviewers’ feedback, we have performed additional experiments accordingly and provided individual responses to each reviewer. The main clarification we made and extra experiments we did are summarized as follows and they will be added to the updated version of the paper:

1.  Making clarification of the terms such as “out-of-distribution”, “unseen tasks”


	1.  In this work, we follow the previous work [1,2,3] and focus on the easy-to-hard generalization. Specifically, the in-distribution/seen tasks here means the testing samples are sampled from the same problem size as the ones shown in the context [3]. For example, we demonstrate examples of 2-digit addition, and then test it over unseen samples that are also from 2-digit addition. In contrast, the out-of-distribution/unseen tasks here are defined to be the harder unseen variants of the problem, which are harder to solve compared to examples provided in the context. For example, the testing samples of 5-digit additions are the harder variant of the problem that are not demonstrated in the context examples (2-digit additions) [3]. And we propose the SKiC prompts to improve such easy-to-hard compositional generalization [3] and complex reasoning tasks compared to previous methods. In some of the tasks, our SKiC actually achieves near-perfect performances on harder cases, which are not demonstrated in the context.


2.  Adding more experimental results, including:


	1.  Adding experimental comparison with previous work (Algorithm prompting) on the tasks of addition, multiplication and GSM8K.

	2.  Adding extra comparison to recent CSV method on MATH dataset.

	3.  Adding experiments where we directly apply the SKiC prompt designed for GSM8K (simple math sets) to the task of MATH (competition math test sets) and FOLIO (logical reasoning) to show the generalization to absolutely unseen tasks. In other words, we show that the SKiC prompt designed for one task can also be directly generalized to a different unseen task.

	4.  Performing new experiments to show the robustness of SKiC prompting to the choice of in-context few-shot exemplars. For example, the performance with different choices of few-shot examplars as well as their ordering.

	5.  Adding ablation studies (on DP tasks), where we remove different components in SKiC to demonstrate the importance of in-context skills and skill grounding in reasoning steps. We show that grounding is the key to improve the easy-to-hard generalization.

	6.  Performing extra experiments on one semantic parsing task to illustrate our SKiC also works for semantic parsing.

	7.  Adding experiments on utilizing LLMs to discover skills from given examples and construct SKiC based on discovered skills for last-letter and general NLU tasks like NLI to show that our SKiC is generally applicable.

	8.  Improving the results on multiplication tasks with the recently released GPT4 with longer context window to show that SKiC could also achieve near-perfect generalization on multiplication tasks besides last letter, addition, and dynamic programming.





[1]Decomposed Prompting: A MODULAR APPROACH FOR SOLVING COMPLEX TASKS

[2]Least-to-Most Prompting Enables Complex Reasoning in Large Language Models

[3] Faith and Fate: Limits of Transformers on Compositionality

---

### Meta-Review · Area_Chair_jKdR · 2023-12-06

**Metareview:**

The authors present a prompting method that explicitly decomposes problems into component skills for some problems in context, and show that this prompt helps performance. Unfortunately this isn't adequately novel or general (compared to CoT, Teaching Algorithmic Reasoning via In-context Learning, etc -- basically other works that decompose an in-context examples into the "right steps" and show performance improvements).

**Justification For Why Not Higher Score:**

isn't adequately novel or general compared to CoT, Teaching Algorithmic Reasoning via In-context Learning, etc

**Justification For Why Not Lower Score:**

na

---

### Decision · Program_Chairs · 2024-01-16

Reject